# Curriculum learning for ab initio deep learned refractive optics

Xinge Yang [1], Qiang Fu [1] & Wolfgang Heidrich [1] ✉

Deep optical optimization has recently emerged as a new paradigm for designing computational imaging systems using only the output image as the objective. However, it has been limited to either simple optical systems consisting of a single element such as a diffractive optical element or metalens, or the fine-tuning of compound lenses from good initial designs. Here we present a DeepLens design method based on curriculum learning, which is able to learn optical designs of compound lenses ab initio from randomly initialized surfaces without human intervention, therefore overcoming the need for a good initial design. We demonstrate the effectiveness of our approach by fully automatically designing both classical imaging lenses and a large field-of-view extended depth-of-field computational lens in a cellphone-style form factor, with highly aspheric surfaces and a short back focal length.

Deep optics has recently emerged as a promising paradigm for jointly optimizing optical designs and downstream image reconstruction methods[1-6]. A deep optics framework is powered by differentiable optical simulators and optimization based on error back-propagation (or reverse mode auto-differentiation) in combination with error metrics that directly measure final reconstructed image quality rather than classical and manually tuned merit functions. As a result, the reconstruction methods (typically in the form of a deep neural network) can be learned at the same time as the optical design parameters through the use of optimization algorithms known from machine learning.

This paradigm has been applied successfully to the design of single-element optical systems composed of a single diffractive optical element (DOE) or metasurface[1,6-11]. It has also been applied to the design of hybrid systems composed of an idealized thin lens combined with a DOE as an encoding element[3,12-17]. In the latter setting, the thin lens is used as an approximate representation of a pre-existing compound lens, while the DOE is designed to encode additional information for specific imaging tasks such as hyperspectral imaging[7-9], extended depth-of-field (EDoF)[1,4,17], high dynamic range imaging[3], sensor super-resolution[1,13], and cloaking of occluders[16].

Most recently, there has been an effort to expand the deep optical design paradigm to compound optical systems composed of multiple refractive optical elements[4-6,18-21]. The core methodology behind these efforts is an optical simulation based on differentiable ray tracing (Fig. 1a), in which the evolution of image quality can be tracked as a function of design parameters such as lens curvatures or placements of lens elements. Unfortunately, this design space is highly non-convex, causing the optimization to get stuck in local minima, a problem that is familiar with classical optical design[22-25]. As a result, existing methods[4,5,26-28] can only fine-tune good starting points, and require constant manual supervision, which is not suitable for the joint design of optics and downstream algorithms. Although there are some works[18,29] for automated lens design, they usually rely on massive training data and therefore fail when design specifications have insufficient reference data. These limitations present a major obstacle to the adoption of deep optical design strategies for real computational optical systems, since the local search around manually crafted initial designs prevents the exploration of other parts of the design spaces that could leverage the full power of joint optical and computational systems.

In this work, we eliminate the need for good starting points and continuous manipulation in the lens design process by introducing an automatic method based on curriculum learning[30-32]. This learning approach allows us to obtain classical optical designs fully automatically from randomly initialized lens geometries, and therefore enables the full power of DeepLens design of compound refractive optics in combination with downstream image reconstruction (Fig. 1a). The curriculum learning approach finds successful optimization paths by initially solving easier imaging tasks, including a smaller aperture size and narrower field-of-view (FoV), and then progressively expanding to more difficult design specifications (Fig. 1d). It also comes with

[1]King Abdullah University of Science and Technology (KAUST), Thuwal, Saudi Arabia. ✉e-mail: wolfgang.heidrich@kaust.edu.sa

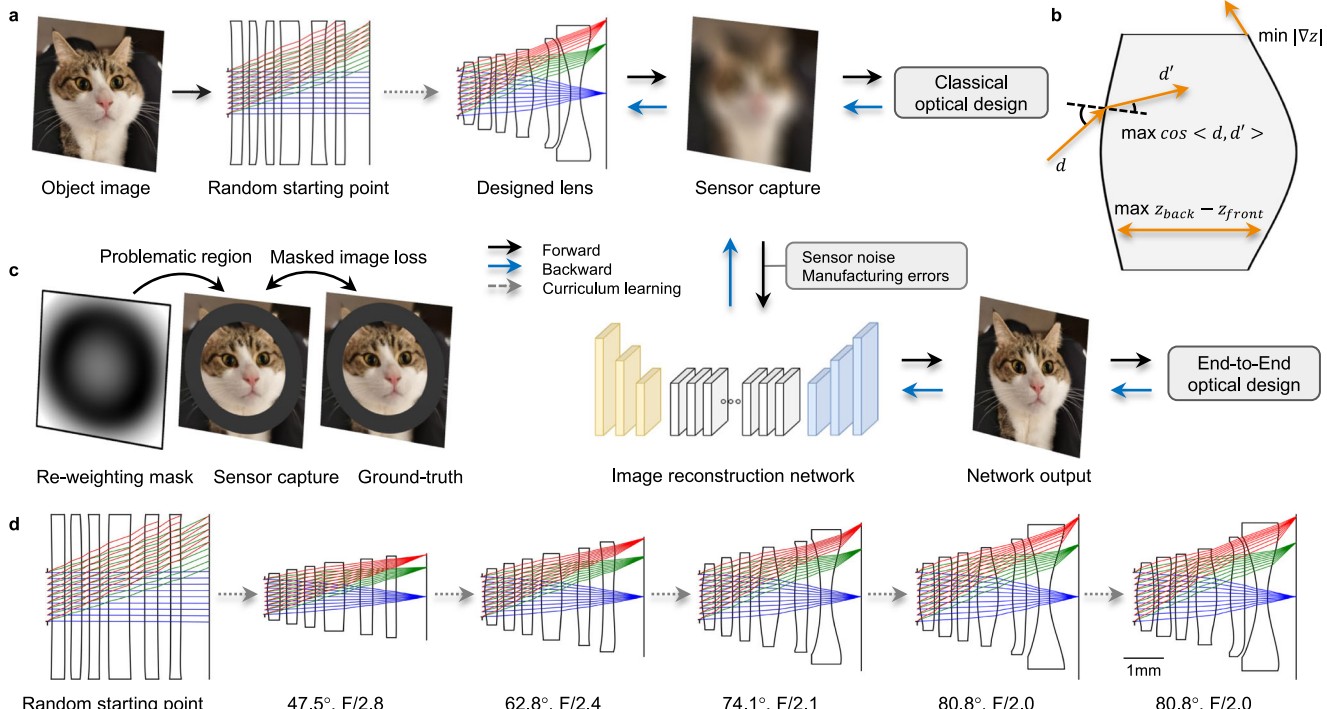

**Fig. 1 | Curriculum learning for automated lens design. a** We utilize a differentiable ray-tracing approach to simulate the sensor captured image of an object image. This sensor capture can then be input into a downstream deep network for image reconstruction. During the forward image simulation (black arrows), we track the gradient of each optical parameter. We can subsequently back-propagate (blue arrows) the errors from either the simulated image for classical optical design, or from the network output for end-to-end optical design. The end-to-end optical design jointly optimizes the optical lens and the image reconstruction network. Classical lens design methods often face issues such as local minima and degenerate optical structures, including self-intersections, requiring appropriate starting points and consistent human intervention. We introduce a curriculum learning strategy that encompasses: a curriculum path (gray dashed arrow in **a**), optical regularization (**b**), and a re-weighting mask (**c**). **b** The optical regularization term presents lens from degenerate structures during the optimization. **c** The re-weighting mask dynamically directs attention towards problematic regions of simulated images during each epoch, compelling the optimization process to escape local minima. This curriculum learning strategy aims to automate the design of complex optical lenses from scratch, for both classical and computational lenses. **d** An example of this automated classical lens design using the curriculum learning strategy. The lens design process initiates from a flat structure, gradually elevating the design complexity until it meets the final design specifications. Detailed evaluations can be found in Table 1 and Supplementary Note 4.1. The cat image was photographed by Xinge Yang (CC BY 2.0).

strategies for effectively avoiding degenerate lens shapes, for example, self-intersection (Fig. 1b), and focusing the optimization on image regions with high error to escape the local minima (Fig. 1c).

To illustrate the power of the framework, we demonstrate its performance and flexibility by designing multiple classical optical lenses with different specifications featuring highly aspherical lens elements and a short back focal length. Furthermore, we showcase a compact EDoF camera design that builds upon common highly aspherical lenses and is complicated by the strong spatial variation of aberrations across the image plane. One of the lens elements incorporates an odd-degree polynomial term, similar to the cubic phase plate used in wavefront coding[33]. This design results in almost depth-invariant PSFs over a large FoV, from which an all-in-focus image can be recovered by the reconstruction network. We believe that the proposed method bridges the gap between optical design and image reconstruction, representing a significant step towards a general framework for any end-to-end DeepLens design application.

## Results

The differentiable ray-tracing framework[4,5] simulates camera-captured images by ray-tracing-based rendering (Fig. 1a). During the forward image simulation, the gradient of each optical parameter $\theta$ (surface curvature, position, conic, and polynomial coefficients) is tracked. Subsequently, the sensor simulation can be input into a downstream deep network $\theta'$ for image reconstruction. In the backpropagation phase, the image error between the object image $I$ and the network

output $\widetilde{I}$ is back-propagated to optimize both optical and network parameters:

$$\theta, \theta' = \arg\min \| \widetilde{I}(I; \theta, \theta') - I \|_2^2. \quad (1)$$

In the subsequent sections, both classical optical design and end-to-end optical design utilize this image-based optimization approach.

## Curriculum learning for automated lens design

Designing complex imaging lenses from scratch presents a highly non-convex problem. This often results in degenerate structures, such as self-intersections and aggressive aspheric shapes, during the optimization process. Additionally, these lens designs can become ensnared in configurations that are locally optimal but globally suboptimal. To enable fully automated lens design, we propose a curriculum learning approach with three key features: a curriculum path that incrementally elevates lens design difficulty, optical regularization to deter degenerate structures, and a spatially re-weighting mask to assist in escaping local minima.

The curriculum learning strategy decomposes the final design target into a sequence of tasks that gradually increase in complexity. This is informed by two well-established observations: 1) geometric optical aberrations are minimized for small apertures, and 2) paraxial regions exhibit fewer aberrations than larger angles. Consequently, the lens design curriculum commences by optimizing the lens for a small aperture and FoV, subsequently expanding both parameters to meet

the final design specifications. Specifically, in our experiments we set the design specifications for each intermediate step as:

$$t_i = t_a + (t_b - t_a) \times \sin\left(\frac{i}{2N}\pi\right), \quad (2)$$

where $t$ represents the design specifications (FoV and F-number), $i$ is the current step, $N$ is the total number of steps, and $t_a$ and $t_b$ are the initial and final design targets, respectively. The sine curriculum path is employed based on the observation that the difficulty of the lens design task increases significantly with larger FoV and aperture sizes.

During the lens design process, optical regularization (Fig. 1b) is employed to prevent degenerate structures such as self-intersections and aggressive aspheric shapes. Additionally, a re-weighting mask (Fig. 1c) is used in each training epoch to dynamically adjust the image-based loss function, thereby concentrating the optimization on challenging regions of the image. Detailed technical aspects of the optical regularization and re-weighting mask can be found in the Methods section.

In Fig. 1d, we showcase an example of the ab initio optimization of a classical lens system without computational post-processing. Starting from nearly planar, randomly initialized lens geometries, our proposed curriculum learning methods successfully design an optical lens with specifications of 80.8°, F/2.0, and a sensor diagonal length of 7.66 mm. We employ a sensor resolution of 2048 × 2048 for mega-pixel imaging; the corresponding pixel size is 2.65 $\mu$m. The initial lens design 47.5°, F/2.8 is not aggressively shaped since the design specifications are relatively moderate. This allows us to directly design it from a randomly initialized structure using differentiable ray tracing. Gradually, the FoV and aperture are increased in stages: first to 62.8°, F/2.4, then to 74.1°, F/2.1, and finally to 80.8°, F/2.0. This is followed by a fine-tuning step to mitigate the geometric distortion of the lens, leading to the final design. This stepwise increase in design complexity enables us to circumvent local minima and degenerate structures, finally yielding the final design. More evaluation of the optical performance is provided in Supplementary Note 4.1.

Further evaluation of the curriculum learning strategy is provided in Table 1. We choose 20 randomly initialized structures for automated lens design, adhering to the design specification (80.8°, F/2.0, 7.66 mm sensor diagonal distance). Specifically, our primary objective is to produce lens structures free from self-intersection. We subsequently compare the average (Avg) and minimum (Min) root-mean-square (RMS) spot size of the final design. The RMS spot size is computed using 256 distinct incident fields to gauge the image performance of the final design. We adopt the basic differentiable ray-tracing method presented in $dO$[5] as our baseline for comparison. For the design specification of 80.8°, F/2.0, the baseline method can not avert self-intersection, resulting in no successful final lens designs. However, by employing the lens design curriculum, which incrementally increases the design FoV and F-number at each step, we achieve 40% of the final designs without self-intersection. Nonetheless, imaging performance encounters local minima, causing blurry areas in the simulated sensor images, also as reflected by the Avg and Min RMS spot size metrics.

When introducing only the optical regularization, all final lens designs successfully sidestep self-intersection, but both the Avg and Min RMS spot sizes witness an uptick. By amalgamating both the curriculum and the regularization term, the automated optimization process produces all lens designs with a reduced Avg RMS spot size of 17.52 $\mu$m and a Min RMS spot size of 14.34 $\mu$m. Lastly, by incorporating the re-weighting mask, the Avg and Min RMS spot sizes further diminish to 15.74 $\mu$m and 12.50 $\mu$m, respectively.

A video animation of the automated optimization process is shown in the accompanying https://youtu.be/32XuSyM-J-8, Supplemental Movie 1. The final design can be further improved by optimizing with more iterations and a higher sensor resolution. More technical details and lens design examples are provided in Supplementary Note 4.

### End-to-end lens design for extended depth-of-field imaging

We also highlight the potential and versatility of curriculum learning by designing a computational camera with EDoF capability using a limited number of highly aspheric elements and a short back focal length. The objective of an EDoF computational camera is to combine the light sensitivity of large apertures with an extensive depth of field. Wavefront coding, as elaborated in prior studies[33–35], involves integrating a cubic phase plate into an optical system to blur the image uniformly in a focus-independent fashion. Following this, a sharp image can be computationally reconstructed by deconvolving the uniformly blurred raw camera capture. However, applying this principle to highly aspheric, wide FoV lenses with a compact structure has posed challenges, primarily because optical aberrations vary substantially across the image plane in such systems.

To address this challenging problem, we choose a design space in which each lens surface is represented by a classical aspheric model that includes spherical, conical, and even polynomial degrees as a function of radial distance $r = \sqrt{x^2 + y^2}$ from the optical axis. Furthermore, one surface in the design permits odd polynomial degrees as a function of $x, y$. The odd polynomials extend the cubic phase plate from wavefront coding to this hybrid surface, which is characterized as:

$$z(r) = \underbrace{\frac{r^2}{R\left(1 + \sqrt{1 - (1+\kappa)r^2/R^2}\right)} + \alpha_2 r^2 + \alpha_4 r^4 + \cdots}_{\text{aspheric}} + \underbrace{\sum_{i=1}^{n}(a_i x^{2i+1} + b_i y^{2i+1})}_{\text{odd-polynomial}}. \quad (3)$$

The hybrid surface introduces additional image blur, but in a controlled manner that facilitates reconstruction of an all-in-focus image. The inspiration for this term is similar to wavefront coding[33], but additional higher orders of odd polynomials are utilized to allow for finer control over the off-axis performance in the presence of real-world aberrations (see Supplementary Note 5.4. and Table S5). An image reconstruction network is necessary as a second component of the computational imaging system. We employed NAFNet[36] as the image reconstruction network which is a UNet[37]-shaped network with optimized inter- and intra-blocks, making it both computationally efficient and easy for training. Additionally, NAFNet demonstrated

### Table 1 | Effectiveness evaluation of the curriculum learning strategy in automated lens design

| | Baseline ($dO$[5]) | + Curriculum | + Reg | + Curriculum & Reg | + Curriculum & Reg & WM |
|---|---|---|---|---|---|
| No self-intersection | 0 | 40% | 100% | 100% | 100% |
| Avg RMS spot size ($\mu$m) | N.A. | 16.70[a] | 22.19 | 17.52 | **15.74** |
| Min RMS spot size ($\mu$m) | N.A. | 13.23[a] | 19.04 | 14.34 | **12.50** |

Smaller RMS spot size indicates better performance. Bold values highlight the best performance. The design target is set to 80.8°, F/2.0, 4.5 mm. 20 random initializations are used to evaluate the results of different curriculum learning strategies. Avg RMS spot size is calculated on 256 distinct fields, for all designs without self-intersection.
*Reg* optical regularization, *WM* re-weighting mask.
[a]Significant ray failure is observed in final designs, falsely reducing the RMS spot size.

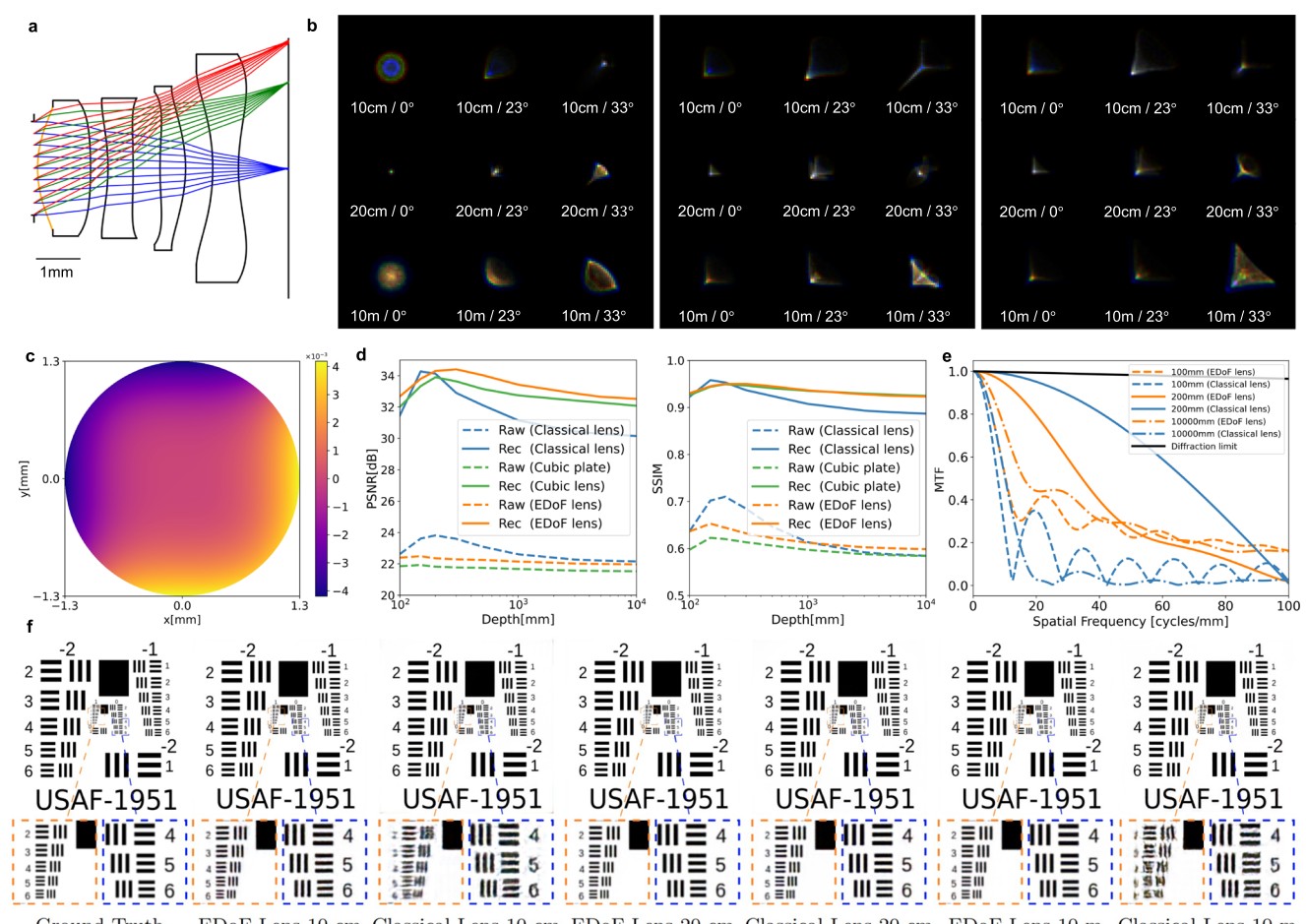

**Fig. 2 | Evaluation of deep learned large-aperture EDoF lens. a** The first surface of a classical aspherical lens is replaced by a hybrid odd-polynomial-aspheric surface to form an EDoF lens. The optical parameters of the EDoF lens are jointly optimized with the image reconstruction network in an end-to-end training manner. **b** For left to right: PSFs of the classical lens, EDoF lens with hybrid surface, and EDoF lens with an extra cubic plate at different depths and view angles. The PSFs of our deep learned EDoF lens are more depth-invariant compared to that of the classical lens.

PSFs of more wavelengths are provided in Supplementary Note 5.3. **c** The height profile of the odd-polynomial term of the hybrid surface, which brings the EDoF ability to the lens system. **d** Image quality evaluation of simulated raw images and network reconstruction with PSNR and SSIM matrices. **e** MTF curves of the classical and EDoF lens at different depths, without image reconstruction. **f** Zoomed patches of network reconstructions at different depths. More evaluation results are provided in Supplementary Note 5.4 and 5.5.

excellent performance on several image deblurring tasks, giving the balance between performance and computational efficiency.

The EDoF lens is designed with large aperture size and a broad depth of field, ensuring clear imaging from 10 cm to 10 m, even under low light conditions. Initially, we design a classical lens with a focal length of 5.55 mm, FoV of 68. 8°, F/2.1 (which corresponds to an aperture diameter of 2.60 mm), and an image height of 7.20 mm. The sensor boasts a resolution of 1024 × 1024, and each pixel is 4.97 $\mu m$ in size. This lens is focused at 20 cm to strike a balance in controlling the defocus blur both at 10 cm and 10 m. Subsequently, we modify the design space by replacing the first surface with a hybrid surface (orange line in Fig. 2a). We will hereafter refer to the proposed design as the "EDoF lens" and the original design as the "Classical lens". We also introduce post-processing of the raw capture using a reconstruction network and jointly optimize the EDoF lens and the network to achieve sharp imaging from 10 cm to 10 m. The loss function is designed as:

$$\mathcal{L} = \sum_{d \neq d'} \mathcal{L}_{\text{sim}}(\widetilde{I}_d, \widetilde{I}_{d'}) + \sum_{d} \left[ \omega_1 \mathcal{L}_{\text{raw}}(\widetilde{I}_d, I) + \omega_2 \mathcal{L}_{\text{rec}}(\bar{I}_d, I) \right], \quad (4)$$

where $I$, $\widetilde{I}$, and $\bar{I}$ represent the object image, simulated raw image, and reconstruction results, respectively. Hyperparameters $\omega_1$ and $\omega_2$

balance different loss terms. At this stage, we optimize all optical and network parameters to learn consistent image simulations across varying depths ($\mathcal{L}_{\text{sim}}$), while also striving for the highest simulation quality ($\mathcal{L}_{\text{raw}}$) and compatibility with the downstream deep network ($\mathcal{L}_{\text{rec}}$). Specifically, we segment the continuous depth range into 8 training depths (10 cm, 15 cm, 20 cm, 30 cm, 50 cm, 1 m, 3 m, 10 m) to mitigate overfitting, and randomly select two of them for each training batch. We use three wavelengths (486 nm, 587 nm, and 656 nm) to simulate different image channels, enabling the network to reduce chromatic aberration. The odd-polynomial term of the deep learned hybrid surface is illustrated in Fig. 2c. In Fig. 2b, the PSFs of the classical lens, EDoF lens with hybrid surface, and EDoF lens with cubic plate at various depths and viewing angles are displayed. PSFs at more wavelengths are provided in Supplementary Note 5.3. The classical lens exhibits significant off-axis optical aberrations, stemming from the demanding design specifications that call for a large FoV and aperture size. Compared to the classical lens, our EDoF lens produces more depth-invariant PSFs. Although the optical aberrations reduce this depth invariance, the subsequent image reconstruction network can adjust for these aberrations and yield a clear output.

In the final stage, following the end-to-end optical design, we fix the optical lens and further refine the network to achieve optimal output quality. Specifically, we simulate the entire dataset across all

training depths and persist with the network training. Additionally, we introduce variations in the optical parameters to model and counter potential challenges stemming from registration inaccuracies in the polynomial-aspheric surface and potential inconsistencies during lens manufacturing. This approach aims to fortify our network's resilience against such deviations, enhancing the robustness of our proposed models. Comprehensive details are provided in Supplementary Note 5.

To evaluate the imaging performance of our EDoF lens, we calculate the average PSNR and SSIM scores on 100 test images at various depths ranging from 10 cm to 10 m. For comparison, we use the classical lens as the baseline and also design another lens by placing a deep-learned cubic plate at the aperture position. Each lens is paired with a reconstruction network of the same architecture. In terms of imaging performance, the classical imaging lens exhibits a significant depth-of-field effect (Fig. 2d, blue dashed line). In contrast, the deep-learned EDoF lens delivers a more uniform imaging performance across a broad depth range (Fig. 2d, orange and green dashed lines), despite the raw captures being blurry. The MTF curves for EDoF and classical lenses without reconstruction at in-focus and out-of-focus depths are shown in Fig. 2e. Our deep-learned EDoF lens demonstrates a more depth-invariant imaging performance at out-of-focus depths in raw captures, compared to the classical lens, aligning with our quantitative evaluation of the testing dataset. A comprehensive comparison between the cubic plate EDoF lens and the hybrid surface EDoF lens is provided in Supplementary Note 5.4.

After post-processing, the reconstruction network improves the image quality for all three lenses. However, it is unable to eliminate the depth-of-field effect in the classical lens (Fig. 2d, blue line), indicating that the reconstruction network is focus-dependent and cannot adequately restore out-of-focus images. Both our EDoF lens and the cubic plate lens enable the network to generate clear images across all depths (Fig. 2d, orange and green lines). However, the cubic plate lens yields inferior output quality, due to the introduction of an additional optical element that causes more optical aberrations. In many real-world scenarios, the inclusion of an extra optical element is typically unfeasible due to space constraints. Zoomed patches of the reconstructed images are shown in Fig. 2f. The results from the deep-learned EDoF lens closely resemble the ground-truth object images across various depths and preserve essential details. In contrast, while the reconstruction results of the classical lens at 20 cm are close to the ground truth, those at 10 cm and 10 m contain significant artifacts, with fine details absent. Quantitative simulation and reconstruction results are provided in Table 2. Our EDoF lens achieves the highest or second-highest reconstruction scores at all three depths (10 cm, 20 cm, and 10 m). Additional evaluations can also be found in Supplementary Figs. S15, S16, and S17.

## Discussion

In mobile camera devices, the image signal processing (ISP) module acts as an intermediary step after the camera hardware captures electronic signals. While the ISP module plays a crucial role in determining image quality for such devices, our current study does not consider its impact, focusing solely on optical design. Two primary reasons underpin this decision. First, narrowing our focus to optical

design offers a clearer context for grasping optical phenomena, especially since EDoF capability is rooted in the geometric features of the optical lens. Second, there is no well-established, common model for ISPs, with strong variations in ISP architecture across devices and public domain alternatives lag significantly behind the technical capabilities of commercial ISP. Moreover, the reconstruction network of the deep lens approach can also be regarded as a stand-in for more sophisticated ISPs, which implements some of the ISP functionality such as sharpening, or color processing, but not other low-level tasks such as de-mosaicking. If a differentiable implementation of an ISP is available, it can either replace or augment the network used in our experiments.

In practical scenarios, the manufacturing process is integral to lens design. However, in this paper, our study primarily focuses on the automated design of optical lenses and does not delve into lens manufacturing. Furthermore, our simulation framework shows similar ray tracing accuracy with commercial software ZEMAX, which is usually esteemed as the industry benchmark for lens design and production. And we adopt ZEMAX for optical performance evaluation for our designed lenses. To address potential challenges in lens manufacturing and assembling, we introduce perturbations to the optical parameters. This anticipates issues like registration inaccuracies in the polynomial-aspherical surface and inconsistencies in lens manufacturing. Such an approach enhances the resilience of our network to deviations, fortifying the robustness of our image reconstruction models.

## Methods
### Differentiable ray tracing
Our DeepLens optimization employs differentiable ray-tracing-based rendering[4,5] for image simulation and back-propagation for optimization. The technical details are provided in Supplementary Note 1. The basic differentiable ray-tracing method presents several challenges, including substantial memory consumption, potential instability, and lens surface self-intersection during optimization. Additionally, the original pixel-wise loss function, Eq. (1), struggles with geometric distortion in lenses with a large FoV. In this work, we introduce several techniques to address these issues.

### Optical regularization
The optical regularization serves to prevent the optimization process from veering into degenerate configurations, such as self-intersecting geometries or aggressive aspheric shapes, which are either physically impractical or not feasible for fabrication. As depicted in Fig. 1b, three regularization losses are employed to penalize the following: 1) the obliquity of the optical rays, formulated as:

$$\mathcal{L}_{\text{angle}} = -\min\left(\sum_{k}^{spp}\prod_{n}^{N}\mathbf{d}_{kn}\cdot\mathbf{d}'_{kn},\epsilon_{\text{angle}}\right), \tag{5}$$

where $spp$ denotes the number of rays sampled from each sensor pixel, and $M$ represents the number of lens surfaces. $\mathbf{d}$ and $\mathbf{d}'$ denote the normalized incident and outgoing ray directions, respectively. This loss function discourages optical rays with large refractive angles,

**Table 2 | Quantitative comparison on different EDoF systems in terms of PSNR/SSIM**

| Method | Classical lens (PSNR/SSIM) | | EDoF lens (Cubic plate) | | EDoF lens (Hybrid surface) | |
|---|---|---|---|---|---|---|
| | simulation | reconstruction | simulation | reconstruction | simulation | reconstruction |
| 10 cm | 22.62/0.637 | 31.46/0.923 | 21.85/0.598 | 32.01/0.926 | 22.40/0.636 | **32.68/0.930** |
| 20 cm | 23.81/0.710 | 34.13/**0.953** | 21.82/0.620 | 33.90/0.949 | 22.35/0.644 | **34.30**/0.950 |
| 10 m | 22.16/0.585 | 30.15/0.887 | 21.53/0.584 | 32.08/**0.925** | 21.97/0.599 | **32.52**/0.923 |

Higher PSNR and SSIM values indicate better performance. Bold values highlight the best performance.

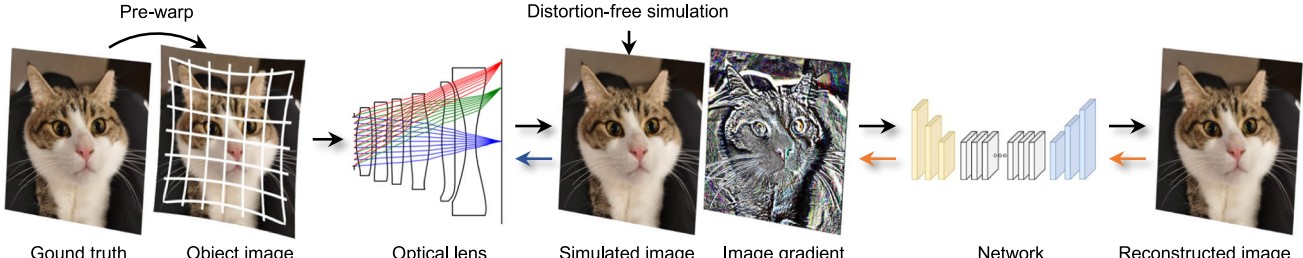

**Fig. 3 | Distortion relaxation and adjoint rendering in differentiable ray tracing.** Distortion relaxation: in large FoV optical length design, geometric distortion is usually relaxed for better control over other optical aberrations. The basic image-based loss function can not address this issue as it calculates per-pixel errors. To allow for geometric distortion in the final designed lens, we calculate the inverse distortion mapping relation and use it to pre-warp the object image for ray-tracing-based rendering. For example, if a lens has barrel distortion, we apply a pincushion distortion to the object image. Then during the image simulation, two distortions cancel out and the simulated sensor image is distortion-free compared to the ground truth. Adjoint rendering: high-resolution differentiable ray tracing consumes a large amount of memory. To address this issue, we propose an adjoint rendering approach that recalculates the ray-tracing simulation during back-propagation. This approach separates the gradient calculation of the network part (orange arrow) and the ray-tracing part (blue arrow), without compromising the calculation. To further improve the efficiency of our approach, we also incorporate a patch backpropagation method that recursively backpropagates gradients for image patches. The cat image was photographed by Xinge Yang (CC BY 2.0).

hereby encouraging a smooth light path. If the function value exceeds an empirical bound $\epsilon_{\text{angle}}$, which is 0.7 in our experiments, the loss function will back-propagate zero gradients to optical parameters and thus not impact the optimization process.

2) the distance between two neighboring surfaces, formulated as:

$$\mathcal{L}_{\text{dist}} = -\min(\delta z, \epsilon_{\text{dist}}), \tag{6}$$

where $\delta z$ denotes the distance between two neighboring surfaces at a specific radial position. This loss function separates two lens surfaces if the distance falls below the threshold $\epsilon_{\text{dist}}$, thereby helping to avoid self-intersections. In our experiments, a set of radial positions on the surface are sampled to calculate the loss function, and the bound $\epsilon_{\text{dist}}$ is set to 0.4 mm for identical lens elements and 0.2 mm for different elements. If the function value is larger than the bound, the loss function will back-propagate zero gradient to optical parameters and thus not impact the optimization process.

3) the surface gradients, formulated as

$$\mathcal{L}_{\text{shape}} = \max\left(\left|\frac{\partial z}{\partial r}\right|, \epsilon_{\text{shape}}\right), \tag{7}$$

where $\frac{\partial z}{\partial r}$ represents the surface gradient at a given radial position $r$. This loss function penalizes the surface gradients that exceed the bound $\epsilon_{\text{shape}}$ to avoid aggressive aspheric shapes. In our experiment, we use the maximum valid surface height to calculate the loss function, and the bound $\epsilon_{\text{shape}}$ is set to 0.5. If the function value is smaller than the bound, the loss function will back-propagate zero gradients to optical parameters and thus not impact the optimization process.

**Re-weighting mask**
The re-weighting mask directs attention towards problematic regions of simulated images during each epoch, compelling the optimization process to escape local minima where the overall image gradient lacks the momentum to escape. The re-weighting mask is designed based on the RMS error to identify problematic regions before each epoch. In our experiments, we trace optical rays from an infinite distance to calculate a 2D RMS error grid. Then we linearly normalize the grid and resize the sensor resolution, and then "drop out" well-optimized regions by setting the mask values to zero for those lower than an RMS error threshold. The threshold is set to 0.8 average RMS error in our experiments. With the proposed optical regularization and re-weighting mask, the image-based loss function is as follows

$$\mathcal{L} = \| M(\widetilde{I} - I)\|_2^2 - \sum_{i=1}^{3} \omega_i \mathcal{L}_i, \tag{8}$$

where $M$ is the re-weighting mask, $\omega_i$ is the weight term for each optical regularization loss, and $\mathcal{L}_i$ is the above three optical regularization terms. In our experiments, we set $\omega_i$ to 0.02 for all three losses. During the lens design, the usage of a re-weighting mask exhibits oscillation to escape the current local minima. However, the optical regularization term ensures that these oscillations do not adversely affect the lens structure and the degenerate structures will be corrected, leading to a successful lens design in the end.

**Distortion relaxation for simulated images**
The normal per-pixel image quality loss in differentiable ray tracing requires good alignment between the reference image and the simulated and reconstructed output. However, at times, we may want to eliminate distortion from the loss function to have better control over other optical aberrations. To achieve this, we estimate the distortion of the current design by tracing the chief rays and then warp the object image to generate a distortion-free sensor simulation. Figure 3c provides an example where, assuming the lens has a barrel distortion, we pre-distort the object with an inverse pincushion distortion, canceling out both distortions during ray-tracing-based rendering. The distortion relaxation approach generates a distortion-free and well-aligned sensor simulation, and also avoids non-differentiable unwarping operations between image simulation and network reconstruction, resulting in smoother back-propagation. To control the amount of permissible distortion, we optionally penalize the magnitude of the alignment error in addition to the image-quality-based loss:

$$\mathcal{L} = \alpha \| \widetilde{I}(I; \theta) - I\|_2^2 + (1 - \alpha) \| \widetilde{I}(\mathcal{F}(I); \theta) - I\|_2^2, \tag{9}$$

where the weight coefficients $\alpha_1$ and $\alpha_2$ are used to balance two terms, controlling the amount of distortion. For example, $\alpha = 1$ optimizes a distortion-free lens, which allows for classical optical designs without computational post-processing.

**Adjoint rendering for memory savings**
A simple implementation of end-to-end training of differentiable ray tracing through automatic differentiation results in prohibitive memory usage. Previous approaches have attempted to address this issue through various means, such as computing adjoint derivatives during the forward pass[38,39], simplifying intermediate computations[5], or using

low sensor resolution and sampling rate[4]. However, these methods have proven inadequate for large-scale, high-resolution deep lens design of complex optical systems.

In this paper, we propose an adjoint rendering approach that recalculates the ray-tracing simulation during back-propagation, similar to the method used in ref. [40]. The basic idea is to manually split the backward pipeline into sequential sub-steps. Our approach first performs non-differentiable ray tracing to simulate the sensor image without tracking gradient information. Subsequently, the gradient calculation of the simulated images is activated, and the images are fed into the reconstruction network and gradients are backpropagated to update the network. An error image is also obtained as a result of the backpropagation. Finally, we reset the pseudo-random number seed to perform an identical differentiable ray tracing and obtain the lens gradients. This adjoint rendering approach separates the gradient calculation of the network part and the ray-tracing part, without compromising the calculation. To further improve the efficiency of our approach, we also incorporate a patch backpropagation method that recursively backpropagates gradients for image patches. The combination of these two approaches reduces memory consumption to an affordable level.

## Data availability

The DIV2K dataset is used in the experiments for both classical lens design and EDoF lens design, which is available at https://data.vision.ee.ethz.ch/cvl/DIV2K/. The processed test images and network checkpoints are available for download at https://zenodo.org/record/8358592.

## Code availability

The DeepLens framework is built on the top of the open-source differentiable ray-tracing engine $dO$[5]. The code for this manuscript is available at https://zenodo.org/record/8358592 under an open-source license permitting not-for-profit research use. Future updates to the code will be published at https://github.com/vccimaging/DeepLens[41].

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

## Acknowledgements

The work was supported by KAUST individual baseline funding (W.H.). We would like to extend our gratitude to Dr. Congli Wang for the insightful discussions and technical support of the *dO* framework at the early stage of this work.

## Author contributions

All authors contributed to the development of the automated lens design method. X.Y. implemented the methods and conducted the experiments. Q.F. provided expertise in lens design. W.H. conceptualized and supervised the project. All authors participated in the writing and review of the manuscript.

## Competing interests

The authors declare no competing interests.
