## [Peer Review File · Nature Communications]

Curriculum Learning for ab initio Deep Learned Refractive OpticsREVIEWER COMMENTS

Reviewer #1 (Remarks to the Author):

Summary of the work

The paper looks at compound lens design, where traditional approaches based on inverse rendering often converge to local minima. To address this, the work suggests the use of "curriculum learning", where the design specifications are progressively made harder; for example, the optimization begins with a smaller aperture and smaller sensor size (or field of view) and progressive increases both. It also suggests a range of ideas including spatial reweighing of the loss function, handling distortions separately. The paper uses this to design a EDOF lensing, with more examples in the supplement.

Strengths

- There are many ideas pertaining to implementation details that I feel are broadly interesting and likely the reason behind some of the successes for this technique. This includes the idea of curriculum learning itself as applied to design of optical systems, the specific way distortion is handled, among others.
- This is clearly a hard problem, one that is central to computational imaging and lens design. So any advance here can have widespread impact.

Weaknesses

I) Lack of comparisons to any previous technique in this space

While I said earlier that this is an important problem, the paper suffers from lack of any comparisons to other compound lens design work. The paper after cites many papers that at least superficially work on the same problem; so it is unclear to me why it doesn't provide direct comparisons of any form.

II) Quality of results are overall lacking in multiple aspects

a) There is one main result presented in the paper. However, it raises a number of questions that need to be resolved.

Figure 1 provides the overview of the curriculum planning approach. What is very surprising to me is the relative small difference in specs between the start of the curriculum and its end. In figure 1, this is 57 to 63 degrees in FOV and $f/3.2$ to $f/2.8$. This does seem all that different in specs. The paper doesn't show any other operating point here, which makes me wonder what the real contributions of the curriculum is. Given the central nature of curriculum learning to the paper (starting with the title) I would have expected a more rigorous evaluation as well as more dramatic changes in parameters.

b) Figure 3 which shows EDOF results is barely depth invariant. As far as depth invariance is concerned, the PSF shows marginal improvements over the traditional lens. So my guess is that the backend DNN that is doing the heavy lifting and cleaning up what is clearly a depth varying PSF. Makes me wonder why the lens fails to produce something more dramatic. Again, the lack of comparisons beyond a standard lens makes this hard to evaluate. What happens if we used a coded aperture system (in the style of Levin's work from the SIGGRAPH 2007) or a lattice focal lens or just placing a bicubic phase plate in the aperture of a camera lens.

c) The paper is also missing any evaluation of how different aspects of its contributions---the curriculum, the weighting scheme, distortion handling---change the solution. Perhaps I missed it, but seems this is an important exploration for this work.

III) Lack of a formal solution

a) The specific form of curriculum learning can be interpreted as a biased version of stochastic sampling, where the spatial and angular spread on the sensor is progressively increased. Clearly such a sampling is biased.

b) Similarly, the use of the reweighting scheme is a heuristic; while it is intuitive, it significantly complicated the interpretability of the solution, not to mention the risk of setting up oscillations in the solution.

c) It wasn't clear to me how the weight mask was designed and used. For example, when were the weights changed? what were the specific weight function used and what was its dependence on error ?

In all, my worry is that the proposed method does not have a strong base in optimization. That is, it is not clear what the actual cost function that is optimizing.

IV) The paper only has simulation results. Assembling a compound lens, one with aspheric components is not for the faint of heart, so perhaps simulation results are ok. But the paper then should ensure its simulation pipeline is as physically-accurate as possible. For example, it wasn't immediately clear to me if, post optimization, the evaluation was done on spectrally-broadband scenes. The training itself was done on a three distinct spectra, which is ok for computational reasons; but it is important to test with broadband sources.

Some minor points

- The second terms in the loss function in Equation (3) needs to be about spatial locations on the sensor plane, not image intensities.

Summary of the review

The paper is an interesting take on lens design. However, as it is currently, the manuscript has too many missing evaluations, both against other work in this space as well as variants of itself. In the absence of such evaluation and their interpretation, it is hard to verify if there are actual contributions here.

Reviewer #2 (Remarks to the Author):

The paper proposes curriculum learning method for deep lens design. Engaging with coarse to fine strategy and the dO engine, they realize an extended depth of field (EDoF) computational camera from a randomly initialized design. The main contribution of this paper is to eliminate the need for good initial designs and overcoming the local minima during lens optimization. Several effective improvements have been applied to existing deep lens design paradigm for compound optical systems.

Overall, this work improves the existing methods and the innovation is not significant. The curriculum learning for automatic lens design is appealing. But the paper is vague on the detailed implementation of chromatic PSF and ISP, which may affect the understanding of beginners in this field and reduce accuracy of image simulation. Moreover, the strategy of curriculum learning seems to cause the optimization to fall into some degenerate configurations (Supplemental Video), which means that the automatic design may not robust enough.

Some concerns that need to be addressed as below.

- (1) The authors select three wavelengths (486 nm, 587 nm, and 656 nm) to simulate different image channels allowing the network to learn to minimize chromatic aberration. As visible imaging covers continuous broadband spectrum, is it accurate enough using only three wavelengths to represent chromatic aberration? Is it possible to use more wavelengths to get a more accurate PSF?
- (2) Details should be given on how to obtain the PSFs in Fig. 3(c) and (d). Do the PSFs consist of incoherent broadband spectrum or single wavelength only?
- (3) It is preferred to see real experiments of the proposed learned camera. Because there is a wide gap between the learned design and real imaging (considering lens fabrication, lens assembling and sensor ISP).
- (4) Due to the non-rotational symmetry of the odd-polynomial surface, lens group need precise registration. Rotation of the odd-polynomial surface may lead to severe image quality degradation. The influences of angular deviation on image quality should be analyzed.
- (5) In supplemental video, there still exist some degenerate configurations, like self-intersection, from Step 2 Iter 12 to 1300. However, the problem is solved at Step3 Iter100. What happens between the two steps in detail?

(6) In supplemental material 3.C, all Fig. 6 should be Fig. S6.

Reviewer #3 (Remarks to the Author):

I review the paper 'Curriculum Learning for ab initio Deep Learned Refractive Optics' submitted by Yang et al. The paper presents a progressive deep learning approach to demonstrate automatic lens design of classical imaging lenses, the paper also extends to work for an extended depth-of-field computational cell phone lens. The key result is the design of EDOF imaging system showing a clear benefit compared to classical imaging system which is a first demonstration of this by deep learning.

The work is clearly valid and the data are solid but I am not sure about the significance of the work in terms of a new and important result to the field.

I have the following general comments:

-Figure 1 shows the 'curriculum' process but we didn't have any detail about the level of performance achieved by the lens design (MTF, spot size....) At each step. Ranging from F3.2 to F2.8 is very modest in terms of lens design, in cell phones we achieved routinely F2.0 and lower. The FOV is also modest. I didn't find any detail about the sensor, number of pixels, chief ray angle constraints. With respect to that, the tools cited in ref 18 (LensNet) is much more interesting and as far as we know, it was not trained with a curriculum approach.

- The EDof imager using 6 lenses is also quite unusual in the field (habitually EDof optics also use to simplify the lens construction). It will be much more interesting to include lens designs implication within the method.

- The author didn't discuss how the method compared to joint optimization which is used to optimize the lens and the image processing simultaneously. I cited the work from Fontbonne (Alice Fontbonne, Hervé Sauer, and François Goudail, "Comparison of methods for end-to-end co-optimization of optical systems and image processing with commercial lens design software," Opt. Express 30, 13556-13571 (2022)).

- End-to-end optimization is quite popular, a recent paper from Princeton (F. Heide) like Côté (to be presented at CPVR 2023) (Côté, Geoffroi, et al. "The Differentiable Lens: Compound Lens Search over Glass Surfaces and Materials for Object Detection." arXiv preprint arXiv:2212.04441 (2022).) Show recent progress on joining design for object detection. Heide works also include ISP tuning of the camera, noise and other artifacts that

may affect the object detection or image reconstruction.

Specific comments:

Line 42: 'mobile form factor' should be detailed as the lens size depend on the field of view, for 70 deg lens, habitually the lens TTL smaller than 3mm (laptop lens).

Line 64-78: The authors must provide the optical performance to assest of the lens performance is comparable to a cell phone lens performance.

Line 79: The weight mask is a method but in lens design habitually we did use fields weight factor, how it compared?

Line 118: Detail about the sensor (pixel pitch, number of pixels, CRA) should be discussed. From the scale (Fig 3), it is hard to figure out which can have a sensor is it.

Line 118: Information about the wavelength coverage and weight according to sensor specs should be added.

Figure 3: The MTF curves should be used rather than the USAF target.

Table 1: It is very hard to understand such a figure of merit. For example, how a 1dB difference will be seen by an observer?

Table 1: Sensor tuning impact as well as noise should be added within the simulation to have a real comparison.

In conclusion, it is hard for me to recommend the publication of the paper in this current form. Particularly, general comments should be addressed and data about the sensor and lens performance must be included.

Author Response to the Manuscript: Curriculum Learning for *ab initio* Deep Learned Refractive Optics

Xingye Yang¹, Qiang Fu¹, and Wolfgang Heidrich^{1,*}

¹King Abdullah University of Science and Technology, Saudi Arabia

*Corresponding author: wolfgang.heidrich@kaust.edu.sa

We sincerely thank all reviewers for their insightful comments and suggestions. We have carefully revised the manuscript according to the comments. Also, we have attached the experimental code and image data for the editor and all the reviewers to reproduce the results. They can be accessed at <https://zenodo.org/record/8358592>. The experimental code contains two parts:

- A automated lens design code with the curriculum learning strategy, using RMS errors. This code finishes the automated lens design process in ~ 6 h on a single 3090 GPU.
- The simulated raw images of the three lenses used in the EDoF experiment, and the corresponding checkpoints for reconstruction network. This code will produce the same image simulation and reconstruction scores as in Table 2.

The following is a detailed point-to-point response to the comments.

Response to Reviewer #1:

Summary of the work:

The paper looks at compound lens design, where traditional approaches based on inverse rendering often converge to local minima. To address this, the work suggests the use of “curriculum learning”, where the design specifications are progressively made harder; for example, the optimization begins with a smaller aperture and smaller sensor size (or field of view), and progressive increases both. It also suggests a range of ideas including spatial reweighing of the loss function, and handling distortions separately. The paper uses this to design an EDoF lens with more examples in the supplement.

Strengths:

- There are many ideas pertaining to implementation details that I feel are broadly interesting and likely the reason behind some of the successes of this technique. This includes the idea of curriculum learning itself as applied to the design of optical systems, and the specific way distortion is handled, among others.
- This is clearly a hard problem, one that is central to computational imaging and lens design. So any advance here can have a widespread impact.

Weaknesses:

I) Lack of comparisons to any previous technique in this space.

While I said earlier that this is an important problem, the paper suffers from a lack of any comparisons to other compound lens design work. The paper cites many papers that at least superficially work on the same problem; so it is unclear to me why it doesn't provide direct comparisons of any form.

[Thank you. To the best of our knowledge, there is not a well-established baseline for comparison in the field of automatic lens design. Notably, we are venturing into the design of highly aspheric lenses, which are typically crafted by experienced optical engineers. Such lens design process usually starts from a good initial structure and requires continuous intervention.

We have referenced various papers encompassing both classical and computational lens design. However, they all rely on successful lens starting points¹⁻⁴ and optimize these for the final design. To date, the sole effort in automatic

lens design comes from Cote et al.’s study⁵. However, when tasked with challenging lens design specifications (e.g., F/2.0, 80°), their system only returns an error such as “There is no match for these specifications. Please adjust the f-number or the half field of view.” This shortcoming stems from their method’s dependence on a vast group of successful designs as training ground-truths. When adequate pre-existing structures are missing, their approach struggles with such specifications. Furthermore, they are limited to designing only spherical lenses. In contrast, designing multi-piece, highly aspheric lenses is substantially more challenging. In our study, we are not tethered to pre-existing lens designs; we consistently commence with random initialization and fine-tune the lens structure to fulfill the given specifications using our proposed curriculum learning method.

To showcase the efficacy of our curriculum learning approach in automated lens design, we have included a comparative experiment in our paper. The dO paper⁴ serves as the comparison baseline. Moreover, we have detailed the success rate of producing non-self-intersecting structures and computed the RMS spot size of the final lens designs based on different experimental configurations.

Please check Main paper Table 1 and lines 91-106, Supplementary Note 4 for details.]

II) Quality of results is overall lacking in multiple aspects.

a) There is one main result presented in the paper. However, it raises a number of questions that need to be resolved. Figure 1 provides an overview of the curriculum planning approach. What is very surprising to me is the relatively small difference in specs between the start of the curriculum and its end. In Figure 1, this is 57 to 63 degrees in FOV and f/3.2 to f/2.8. This does seem all that different in specs. The paper doesn’t show any other operating point here, which makes me wonder what the real contributions of the curriculum are. Given the central nature of curriculum learning to the paper (starting with the title), I would have expected a more rigorous evaluation as well as more dramatic changes in parameters.

[Thank you for your feedback. To address your concerns, we have updated the content by replacing the original lens design example with a new design that features an F-number of 2.0 and a diagonal FoV of 80.8°, as depicted in Figure 2. The new design example starts from the specifications of F/2.8 and a diagonal FoV of 47.5°.

Additionally, we have added a more rigorous evaluation in Table 1. The basic differentiable ray tracing method presented in the dO⁴ paper fails to produce non-self-intersecting structures. By adding optical regularization into the optimization, we can successfully avoid degenerated lens structures like self-intersection. By applying the curriculum learning strategy and a dynamic weight mask, we can successfully optimize the lens structure to minimize the RMS spot errors, including jumping out from local minima during the lens design process. Moreover, all final designs are included in the supplementary material.

Please check the Main paper Table 1, lines 80-90, and Supplemental Note 4 for details.]

b) Figure 3 which shows EDoF results is barely depth invariant. As far as depth invariance is concerned, the PSF shows marginal improvements over the traditional lens. So my guess is that the backend DNN is doing the heavy lifting and cleaning up what is clearly a depth-varying PSF. Makes me wonder why the lens fails to produce something more dramatic. Again, the lack of comparisons beyond a standard lens makes this hard to evaluate. What happens if we use a coded aperture system (in the style of Levin’s work from the SIGGRAPH 2007) a lattice focal lens or just place a bicubic phase plate in the aperture of a camera lens?

[Thank you for the comment. We note that most existing EDoF work employs PSF engineering with highly idealized models of the remaining optical system – i.e. the rest of the lens is modeled as an ideal diffraction limited lens. Wavefront coding for example is designed only for on-axis PSFs of a diffraction limited lens, which may be appropriate for microscopy and similar settings but does not accurately model most regular camera lenses, explaining the lack of products in this space. The mentioned work by Levin et al. models neither the diffraction in the mask pattern nor the lens aberrations, so that the final deconvolution had to be performed with a calibrated PSF rather than the designed one.

The contribution of our work is to be able to accurately model *real world lens aberrations*, and design EDoF encodings accordingly, and for a wide field of view! To our knowledge this is not possible with any existing method. We also note that for most camera applications, extra features like EDoF can only be deployed if there no noticeable degradation in image quality compared to a conventional lens system; this explains why we have initially opted for a more subtle DoF expansion.

The PSF in Figure 3 is “barely” depth invariant arises due to two main reasons: (1) optical aberrations diminish the depth invariance. (2) getting an ideal depth invariant PSF usually increases the PSF size too much, which is not optimal for the image processing network. Actually “we optimize all optical and network parameters to learn consistent image simulations across varying depths (\mathcal{L}_{sim}), while also striving for the highest simulation quality (\mathcal{L}_{simu}) and compatibility with the downstream deep network (\mathcal{L}_{rec}).” (Main paper lines 153 - 156.) We agree with

your point that “the backend DNN that is doing the heavy lifting”, and in the revised submission, we have added a new EDoF lens example (4P, F/2.1, diagonal FoV 68.8°) which exhibits more dramatic PSF shifts with varying depth and FoV. Also, we extended the imaging depth range from 10cm to 10m, previously from 20cm to 10m. The image reconstruction results demonstrate that the downstream DNN can successfully correct the blur of the EDoF lens at the existing optical aberrations.

In our original submission, we compared our work with a bicubic phase plate positioned at the aperture. Detailed design and assessment outcomes can be found in the supplemental material. In the revision, we have moved more evaluations to the main paper. (Main paper Table 2, Fig. 2e,f, lines 157 - 178, Supplemental Note 5.)

A comparison with coded-aperture techniques is not feasible, primarily because the coded-aperture plate is typically modeled using wave optics, whereas we employ ray tracing. Currently, there remains a gap between the ray tracing and wave propagation models, which prevents us from simulating wave plates and optical aberrations simultaneously. We are working on a novel method that simultaneously simulates both optical aberrations and diffractive phenomena, which will be presented in a future paper.]

c) The paper is also missing any evaluation of how different aspects of its contributions—the curriculum, the weighting scheme, distortion handling—change the solution. Perhaps I missed it, but seems this is an important exploration for this work.

[Thank you. We have added a more rigorous evaluation in the revised submission. Please check Main paper Table 1 and lines 91 - 106, Supplemental Note 4 for details.]

III) Lack of a formal solution.

a) The specific form of curriculum learning can be interpreted as a biased version of stochastic sampling, where the spatial and angular spread on the sensor is progressively increased. Clearly, such a sampling is biased.

[Thank you. In the context of non-convex optimization problems, such as deep learning and lens design, there is not yet a successful theoretical proof for unbiased global searching. Notably, in lens design, optical engineers often manually steer the optimization process to sidestep local minima, which is inherently biased and subjective.

It is also important to underscore that our objective is not necessarily to find a global minimum. Instead, we aim to identify a viable learning trajectory and arrive at an effective lens design. To the best of our knowledge, that has not been accomplished previously. Moreover, we can also employ various random seeds to expand our search space, thereby mitigating the effects of biased sampling.]

b) Similarly, the use of the reweighting scheme is a heuristic; while it is intuitive, it significantly complicates the interpretability of the solution, not to mention the risk of setting up oscillations in the solution.

[Thank you. We agree with your point that the re-weighting scheme introduces oscillations into the solution. However, these oscillations are intentional. We aim to leverage them in the optimization process to escape local minima. similar to the overshoots produced by momentum-based optimization methods. Additionally, the optical regularization term ensures that these oscillations do not adversely affect the lens structure. As the design nears completion, the influence of these oscillations diminishes since different image regions become more uniformly optimized.

Please check the Main paper “Methods” section, lines 209 - 244, and Supplementary Note 4.2 for details.]

c) It wasn’t clear to me how the weight mask was designed and used. For example, when were the weights changed? What was the specific weight function used and what was its dependence on error?

[Thank you. In the supplemental material, we provide a detailed explanation of the calculation and usage of the re-weighting mask. In the revised submission, we have incorporated more specifics about the re-weighting mask into the main paper.

Please check the Main paper “Methods” section, lines 232 - 244, and Supplementary Note 4.2 for details.]

In all, my worry is that the proposed method does not strong base in optimization. That is, it is not clear what the actual cost function that is optimizing.

[Thank you. It is true that there is no elaborate theoretical framework for our curriculum learning. However, the idea of a training curriculum is well established in other fields with highly non-convex objectives (e.g. robotics and animation). Like in other aspects of machine learning the theory lags behind the state of the art in algorithms, but this is the price to pay for achieving progress on very hard problems. The actual cost function we are optimizing is provided in Eq. (8) of the revised manuscript.]

IV) The paper only has simulation results.

Assembling a compound lens, one with aspheric components is not for the faint of heart, so perhaps simulation results are ok. But the paper then should ensure its simulation pipeline is as physically accurate as possible. For example, it wasn't immediately clear to me if, post optimization, the evaluation was done on spectrally-broadband scenes. The training itself was done on three distinct spectra, which is okay for computational reasons; but it is important to test with broadband sources.

[Thank you. At this moment, the cost of manufacturing a lens is beyond the budget of us. Nevertheless, the accuracy of our simulations aligns closely with the commercial software Zemax, which often serves as an industry-standard in many contexts. Indeed after the design process, the results are validated in Zemax, which was also used to produce the ray diagrams in the manuscript. As presented in the supplementary material, we upload our designs into Zemax and assess their optical performance. Notably, the outcomes from both our simulations and Zemax are consistent.

We only take three wavelengths into account, mirroring the RGB sensors commonly in use. Another reason is the lack of a full-spectra image dataset for network training. In the supplemental material, we have included an evaluation of the PSF of two lenses with more wavelengths.

Please check the Main paper "Discussion" section, lines 194 - 201, Supplementary Note 4 and S5 for details.]

Some minor points:

- The second term in the loss function in Equation (3) needs to be about spatial locations on the sensor plane, not image intensities.

[Thank you. After double-checking we are sure the Eq. (3) (Eq. (9) in the revised submission) is correct. The spatial locations are encoded in the pre-warped object images $\mathcal{F}(I)$, therefore we can directly work on image intensities.]

Summary of the review:

The paper is an interesting take on lens design. However, as it is currently, the manuscript has too many missing evaluations, both against other work in this space as well as variants of itself. In the absence of such evaluation and their interpretation, it is hard to verify if there are actual contributions here.

Response to Reviewer #2:

Overview:

The paper proposes a curriculum learning method for deep lens design. Engaging with coarse to fine strategy and the dO engine, they realize an extended depth of field (EDoF) computational camera from a randomly initialized design. The main contribution of this paper is to eliminate the need for good initial designs and overcome the local minima during lens optimization. Several effective improvements have been applied to existing deep lens design paradigms for compound optical systems.

Overall, this work improves the existing methods and the innovation is not significant. The curriculum learning for automatic lens design is appealing. However, the paper is vague on the detailed implementation of chromatic PSF and ISP, which may affect the understanding of beginners in this field and reduce the accuracy of image simulation.

[Thank you. We believe that our work offers significant impact and innovation in both classical lens design and computational lens design. To the best of our knowledge, no other research has addressed fully automated lens design, especially for compound high-aspheric lenses. At present, lens design is a time-consuming and labor-intensive process that demands substantial human expertise, but our work automates this lens design process.

We would like to stress that we see this technology as a key enabler to bring the “Deep Lens” concept – end-to-end learned optical systems with matching reconstruction methods – from single element optics like DOEs or metalenses to complex optical systems. This can only be successful if real optical systems with real aberrations can be trained end-to-end from arbitrary initializations, which has not been possible before.

In our paper, we employ differentiable ray tracing with RGB wavelengths for image simulation instead of using PSF convolution; therefore, we do not engage in PSF calculation. Nevertheless, deriving the PSF from a spot diagram generated by ray tracing is a well-established method, and it has been discussed extensively in the literature as well as in our previous publications^{4,6,7}. Moreover, the principles and technical details of PSF calculating are very similar to our differentiable ray tracing method. However, in the revised version, we have incorporated the implementation of chromatic PSF calculation in the supplemental material (Supplemental Document 1 Sec S3.5). We also acknowledge that the monochromatic PSF in Fig.2 (c) and (d) may be misleading. Consequently, in the revised manuscript, we have updated Fig.2 (c) and (d) to feature a chromatic PSF in place of the monochromatic one.

We did not consider the ISP for two reasons: (1) There is no well-established differentiable ISP model. Each camera possesses its own ISP, and these differ across companies. Implementing the ISP poses no challenge in principle if we are aware of the specific ISP in use. (2) We believe that omitting the ISP provides a more conducive setting for End-to-End learning. The novel EDoF characteristic of the lenses stems primarily from the optical encoding and not from the ISP module. By overlooking the ISP, end-to-end training can concentrate more on learning the optics. In practice, we can fine-tune the downstream network to counteract the effects of the ISP module, whereas the optical characteristics of the camera lens remain constant. We have included a discussion about this in the revised manuscript.

Please check the Main paper “Discussion” section, lines 180 - 191 for details.]

Moreover, the strategy of curriculum learning seems to cause the optimization to fall into some degenerate configurations (Supplemental Video), which means that the automatic design may not be robust enough.

[Thank you. I guess there is a misunderstanding regarding our supplemental video and curriculum learning strategy. The “oscillation” is, in fact, a crucial feature of our curriculum learning strategy. We encourage the lens design process to exhibit oscillation to escape the current local minima. Additionally, the optical regularization term ensures that these oscillations do not adversely affect the lens structure and the degenerated structures will be corrected, leading to a successful lens design in the end. Please refer to both Fig.1 and Table 1 for an evaluation of the success rate and imaging performance. A corresponding discussion has also been added to the revised manuscript.

Please check the Main paper “Methods” section, lines 209 - 244, and Supplementary Note 4.2 for more details.]

Concerns:

Some concerns that need to be addressed as below.

(1) The authors select three wavelengths (486 nm, 587 nm, and 656 nm) to simulate different image channels allowing the network to learn to minimize chromatic aberration. As visible imaging covers a continuous broadband spectrum, is it accurate enough to use only three wavelengths to represent chromatic aberration? Is it possible to use more wavelengths to get a more accurate PSF?

[Thank you. Indeed, the visible imaging spectrum covers a broadband spectrum, and it is certainly possible to consider more wavelengths. However, we chose these three wavelengths (486 nm, 587 nm, and 656 nm) to represent the RGB channels, which is a commonly accepted practice in both classical optical design and computational

imaging. This approach is reflected in commercial software like ZEMAX where three wavelengths are often enough to characterize RGB channels.

In the revised manuscript, we have included an evaluation of PSF with more wavelengths. The PSFs at different wavelengths change slowly and smoothly, making it reasonable to represent the full spectrum by only three wavelengths.

Please check the Supplementary Note 5.3 for more details.]

(2) Details should be given on how to obtain the PSFs in Fig.3(c) and (d). Do the PSFs consist of an incoherent broadband spectrum or a single wavelength only?

[Thank you for pointing that out. The PSFs in Fig.2(c) and (d) are derived from incoherent ray tracing. In the original manuscript, we only used a single wavelength for PSF calculation, although we took into consideration the RGB three wavelengths. To provide a clearer representation, we have now updated Fig.2 (c) and (d) to showcase a chromatic PSF. We initially omitted an explanation of PSF calculation in the manuscript, as this method is well-established and extensively discussed in the literature, including in our previous publications. In the revised version, we have incorporated the implementation of chromatic PSF calculation in the supplemental material.

Please check Supplementary Note 3.5 and 5.3 for details.]

(3) It is preferred to see real experiments of the proposed learned camera. Because there is a wide gap between the learned design and real imaging (considering lens fabrication, lens assembling, and sensor ISP).

[Thank you for emphasizing the importance of real-world experiments. We fully understand and appreciate the potential gap between simulation and actual implementation, especially in the realm of lens design. Currently, we are actively exploring collaborations with lens manufacturers. However, the financial cost of actual lens fabrication remains a constraint, which is why real experiments were not presented in this paper.

Nevertheless, we have optimized our simulations to align closely with ZEMAX, which usually serves as an industry standard for lens design. To validate our final designs, we imported them into ZEMAX for optical performance evaluation. In response to your feedback, we have included a more comprehensive discussion on this matter in the revised manuscript.

Please check the Main paper “Discussion” section, lines 193 - 201, and Supplementary Note 5.2 for details.]

(4) Due to the non-rotational symmetry of the odd-polynomial surface, the lens group needs precise registration. Rotation of the odd-polynomial surface may lead to severe image quality degradation. The influences of angular deviation on image quality should be analyzed.

[Thank you for highlighting the importance of considering the effects of angular deviation. We agree with your point that any misalignment or rotation of the odd-polynomial surface can potentially compromise image quality. To simulate and address potential issues arising from registration errors, as well as from lens manufacturing discrepancies, we introduced perturbations to the optical parameters during the final stages of network fine-tuning in the revised manuscript. This approach aims to train our network to be more resilient to such deviations, thereby enhancing the robustness of our proposed models. We have included a discussion about this in the revised manuscript.

Please check the Main paper “Results” section, lines 153 - 156, “Discussion” section, lines 193 - 200, and Supplementary Note 5.2 for details.]

(5) In the supplemental video, there still exist some degenerate configurations, like self-intersection, from Step 2 Iter 12 to 1300. However, the problem is solved in Step3 Iter100. What happens between the two steps in detail?

[Thank you for your keen observation regarding the supplemental video. Between Step 2 Iter 12 to 1300, the optimization process exhibits some degenerate configurations, which is a phenomenon of “oscillation” mentioned before. However, the optical regularization loss helps mitigate such degeneracies and results in an improved configuration. A corresponding discussion has been added to the revised manuscript.

Please check the Main paper “Methods” section, lines 209 - 244 for details.]

(6) In supplemental material 3.C, all Fig.6 should be Fig.S6.

[Thanks. We have fixed it in the revised manuscript.]

Response to Reviewer #3:

Overview:

I reviewed the paper “Curriculum Learning for ab initio Deep Learned Refractive Optics” submitted by Yang et al. The paper presents a progressive deep learning approach to demonstrate the automatic lens design of classical imaging lenses, the paper also extends to work for an extended depth-of-field computational cell phone lens. The key result is the design of the EDOF imaging system showing a clear benefit compared to the classical imaging system which is a first demonstration of this by deep learning. The work is clearly valid and the data are solid but I am not sure about the significance of the work in terms of a new and important result to the field.

General comments:

I have the following general comments:

-Figure 1 shows the “curriculum” process but we didn’t have any detail about the level of performance achieved by the lens design (MTF, spot size. . . .) at each step.

[Thank you for your feedback. We did not evaluate the optical performance of the intermediate steps, as they are not our final design objectives. Moreover, a perfect intermediate design might lead to local minima which could hinder achieving our ultimate goal. Our primary focus was ensuring the optimization process progresses and overcomes local minima. In the revised submission, we have included the optical performance evaluation of each intermediate step in the supplemental material.

Please check Supplementary Note 4.1 for details.]

Ranging from F3.2 to F2.8 is very modest in terms of lens design, in cell phones we achieved routinely F2.0 and lower.

[Thank you for pointing that out. We have conducted a new experiment using a lens design with an F-number of 2.0 and a FoV of 80.8°. Depicted in Fig.1(d), the new design has a 35-mm equivalent focal length of 25 mm, aligning closely with the main camera specifications of contemporary cellphones. To further ensure robustness, we also executed 20 tests using different random seeds to gauge the success rate of this design. The results are presented in Table 1 and the figures are in the supplemental material.

Please check Main paper Table 1, lines 80 - 90, and Supplementary Note 4.1 and 4.2 for details.]

The FOV is also modest. I didn’t find any detail about the sensor, number of pixels, or chief ray angle constraints.

[Thank you. This information was mentioned in the supplemental material in the original submission. We have moved these details to the main paper in the revised submission.

Please check Main paper lines 84 and 132 for details.]

With respect to that, the tools cited in ref 18 (LensNet) are much more interesting, and as far as we know, it was not trained with a curriculum approach.

[Thank you. There are substantial differences between our work and LensNet:

(1) While LensNet offers certain capabilities, there are challenges in using it for complex design specifications, such as short focal length, small F-number, and large FoV. For instance, when one inputs specifications like FoV 80.8° and F/2.0 with any focal length, the tool only returns a message “There is no match for these specifications. Try to change the f-number or half field of view.”, this indicates the lack of matching designs.

(2) The limitations with LensNet can be attributed to its dependence on supervised learning, necessitating a vast pool of training data. However, many design specifications might not have accessible data. In contrast, our approach employs unsupervised learning, eliminating the need for specific training data. Our method leverages curriculum learning strategies during the optimization process, ensuring a smooth learning trajectory and avoidance of local minima. This strategy broadens our capability to address more complex lens design specifications.

(3) Additionally, it is worth noting that LensNet can only design spherical lenses. Our focus on designing multi-piece highly aspheric lenses presents an inherently more challenging task.]

- The EDOF imager using 6 lenses is also quite unusual in the field (habitually EDOF optics are also used to simplify the lens construction). It will be much more interesting to include lens design implications within the method.

[Thank you for pointing out this. Acknowledging the industry norms and the significance of simpler constructions, we have taken your suggestion into account. In the revised version, we have incorporated a design for a 4P EDOF lens with specifications of FoV 68.8°, F/2.1, and a 7.2mm sensor.

Please check Main paper lines 129 - 136 for details.]

- The author didn't discuss how the method compared to joint optimization which is used to optimize the lens and the image processing simultaneously. I cited the work from Fontbonne (Alice Fontbonne, Hervé Sauer, and François Goudail, "Comparison of methods for end-to-end co-optimization of optical systems and image processing with commercial lens design software," *Opt. Express* 30, 13556-13571 (2022)).

[Thank you.

It is pertinent to mention that our EDoF lens design is indeed jointly optimized with the image processing network. This approach has been detailed in the "Introduction" section, where we have cited and discussed our methodology in the context of various End-to-End optical design works. Our contribution stands out as we present, for the first time, the design of a multi-piece high-aspheric lens from scratch.

The paper you have referenced specifically uses a simple Cooke triplet lens for their End-to-End training, which streamlines the lens design challenge. Moreover, it is not feasible to compare the performance of two lenses with totally different specifications, the Cooke triplet lens has a FoV of 40°, F/4, 50mm focal length, while our lens has a FoV of 68.8°, F/2.1. 5.5mm focal length.

While the comparison with other End-to-End optical design techniques is valuable, it was not the primary focus of our work, especially since prior works typically only consider on-axis performance and/or PSF engineering for otherwise diffraction limited optical systems, whereas our design considers real aberrations for wide field of view imaging. Moreover, our unique contribution is the ability to obviate the need for a good starting point, commonly required in previous works. Our fully automated lens design approach, we believe, holds significant implications for both classical and computational lens design.]

- End-to-end optimization is quite popular, a recent paper from Princeton (F. Heide) like Côté (to be presented at CPVR 2023) (Côté, Geoffroi, et al. "The Differentiable Lens: Compound Lens Search over Glass Surfaces and Materials for Object Detection." arXiv preprint arXiv:2212.04441 (2022).) Show recent progress on joining design for object detection. Heide's works also include ISP tuning of the camera, noise, and other artifacts that may affect object detection or image reconstruction.

[Thank you for bringing up recent advancements in the field and mentioning the works of Côté. We did not cite this paper because it came out after our original submission. We have added the corresponding reference in the revision.

In this study, we specifically choose to focus solely on the optical design, steering clear of incorporating the ISP module for the following reasons:

(1) The differentiable ISP models are generally not well-established. If the ISP model for the end device is known, there is no challenge in principle to embed it into our framework. However, it often remains a black box, varying across different devices.

(2) Concentrating on the optical design enables a detailed exploration of the optical phenomena associated with camera lenses, especially delving into the nuances of optical aberrations that predominantly stem from geometric optics principles. We have taken into account sensor noise in our experiment, but only in the network fine-tuning phase. We refrained from considering it, as well as ISP tuning and manufacturing artifacts, as they typically fall outside the purview of the lens design phase. The valid lens design gradients come from the optics and network part, instead of sensor noise and other artifacts, which only perturbs the optimization by providing biased and incorrect gradients.

Please check the Main paper "Discussion" section, lines 181 - 191 for more details.]

Specific comments:

Line 42: "mobile form factor" should be detailed as the lens size depends on the field of view, for 70 deg lens, habitually the lens TTL is smaller than 3mm (laptop lens).

[Thank you for pointing this out. We have changed our description in the paper.]

Line 64-78: The authors must provide the optical performance to assess if the lens performance is comparable to a cell phone lens performance.

[Thank you. Please check our supplemental materials for the optical performance evaluation of our automatically designed lenses (Supplementary Note 4). Due to the length limit of the paper, all technical details and optical evaluations are moved to the supplemental material.]

Line 79: The weight mask is a method but in lens design, we did use the field weight factor, how does it compare?

[Thank you for pointing out this. In our approach, the weight mask is a dynamic 2D weight term, which is updated at each training epoch throughout the optimization process. This provides adaptability and directs attention towards problematic regions of simulated images during optimization. In contrast, traditional software like ZEMAX typically employs a static weight factor applied to optical rays from varying incident angles, without the capability for dynamic adjustments during optimization.

Please check the Main paper “Methods” section, lines 232 - 244 for details.]

Line 118: Details about the sensor (pixel pitch, number of pixels, CRA) should be discussed. From the scale (Fig 3), it is hard to figure out which can have a sensor, is it?

[Thank you. We have included this information in the revised submission. Please check Main paper lines 84, and 132 for details.]

Line 118: Information about the wavelength coverage and weight according to sensor specs should be added.

[Thank you. In our experiments, we use an RGB sensor, also because of the RGB image datasets. Three wavelengths (486 nm, 587 nm, and 656 nm) are used to simulate different image channels. A more detailed evaluation of PSFs at different wavelengths is included in the supplemental material.

Please check Supplementary Note 5.3 for details.]

Figure 3: The MTF curves should be used rather than the USAF target.

[Thank you. In the original submission, we used the USAF target to provide a qualitative evaluation of the final output images. In the revision, we have added the corresponding MTF figure at different depths in Fig (2).

Please check Main paper Fig. 2f for details.]

Table 1: It is very hard to understand such a figure of merit. For example, how a 1dB difference will be seen by an observer?

[Thank you for raising this concern about the clarity of the figure of merit in Table 1. PSNR and SSIM are widely recognized and employed as metrics for image quality assessment, especially in deep learning and image processing. While they provide a quantitative assessment, it might not always translate directly to perceptual differences to an observer. To offer a clearer sense of the practical implications of these metrics, we have provided visual reconstructions in Fig. 2g and the supplementary figures Fig. S21, S22, and S23. These visualizations aim to give a qualitative insight into the differences that various PSNR scores might manifest, making it easier to appreciate the variations in image quality.]

Table 1: Sensor tuning impact as well as noise should be added within the simulation to have a real comparison.

[Thank you for pointing this out. In our original submission, we accounted for sensor noise but did not consider the impact of sensor tuning. We chose not to include it because it typically falls outside the purview of the lens design phase. In the network fine-tuning phase, we agree that considering all these factors for a more accurate simulation is important. However, this is not the primary focus of our paper. Our contribution in this paper is to propose, for the first time, a fully automated design method for both classical and computational lenses. We have included a more comprehensive discussion on this choice in the “Discussion” section, lines 181 - 191, of the main paper, as well as in Supplementary Note 5.1.]

Conclusion:

In conclusion, it is hard for me to recommend the publication of the paper in this current form. Particularly, general comments should be addressed and data about the sensor and lens performance must be included.

References

1. Tseng, E. *et al.* Differentiable compound optics and processing pipeline optimization for end-to-end camera design. *ACM Trans. Graph.* **40**, 1–19 (2021).
2. Côté, G., Mannan, F., Thibault, S., Lalonde, J.-F. & Heide, F. The differentiable lens: Compound lens search over glass surfaces and materials for object detection. In *Proceedings of the IEEE/CVF Conference on Computer Vision and Pattern Recognition*, 20803–20812 (2023).
3. Sun, Q., Wang, C., Qiang, F., Xiong, D. & Wolfgang, H. End-to-end complex lens design with differentiable ray tracing. *ACM Trans. Graph.* **40**, 1–13 (2021).

4. Wang, C., Chen, N. & Heidrich, W. *dO*: A differentiable engine for Deep Lens design of computational imaging systems. *IEEE Trans. Comput. Imaging* (2022).
5. Côté, G., Lalonde, J.-F. & Thibault, S. Deep learning-enabled framework for automatic lens design starting point generation. *Opt. Express*. **29**, 3841–3854 (2021).
6. Yang, X., Fu, Q., Elhoseiny, M. & Heidrich, W. Aberration-aware depth-from-focus. *IEEE Transactions on Pattern Analysis Mach. Intell.* (2023).
7. Peng, Y. *et al.* Learned large field-of-view imaging with thin-plate optics. *ACM Trans. Graph.* **38**, 219–1 (2019).

REVIEWER COMMENTS

Reviewer #1 (Remarks to the Author):

The revised paper is significantly improved. The paper now shows results on designs that are significantly different between the starting design specs and the final ones.

The paper also addressed many of the minor comments I had. Overall, I am for accepting this manuscript.

Reading the paper again, the paper would benefit from expanding upon the results in Figure 2 / Table 2

Fig 2 results look a lot like the bicubic phase plate [Cathey and Dowski, 1997]. Could the paper comment on differences between the two? For example, how close is the optimized mask to a bicubic one? How does the depth invariance change across the field? The paper claims in Line 117 that the cubic phase plate has limitations due to "optical aberrations vary substantially across the image plane in such systems". Can this be established by expanding on the results in Table 2, which currently only shows a marginal difference between the cubic plate and the optimized one.

Also, a typo in Fig 2 caption: "depp" to "deep"

Reviewer #2 (Remarks to the Author):

Overview :

After reading the reviews, response, and the revised paper, I appreciate the authors' efforts in rewriting the paper and addressing all the raised issues by the reviewers.

I support the authors' work on the optical optimization of compound high-aspheric lenses from random initio. With three optical regularizations, the authors deploy numerous automated design experiments and alleviate 40% of the final designs from the panic of self-intersection. This significantly decreases the reliance on a well-structured initial design in

the optimization of compound complex lens group. I am particularly thankful for the authors' open-source code, as their approach has the potential to make a significant impact on the broader community.

I agree with authors' discussion about ISP and chromatic PSF. However, when it comes to analyzing the effects of manufacturing and assembly errors, the authors mention fine-tuning based on perturbations but don't provide corresponding simulation experiment data to validate this point.

Afterall, there are still some concerns:

1. I cannot find any implementation details about the cubic phase plate in the article or in the open-source code. It appears to be deployed at the entrance pupil, but I'm unclear about how it introduces phase to the rays in this work.
2. In Fig2.e, the imaging quality of classical lenses, as measured by the Raw PSNR, has degraded approximately 3db compared to the first version of the article.
3. A reference line for the diffraction limit needs to be included in the MTF chart in Fig2.f.
4. In Fig2.g, the simulated results of the classical lens at 10 cm and 10 m have noticeable artifacts that are distinct from typical degradation caused by optical aberrations.
5. Classical lens should be further optimized, or alternatively, patent lens with the similar specifications can be employed to fairly evaluate the results of deep learned large-aperture EDoF lens.

Some minor points:

1. Equation 2: the last term should be $\sin(i\pi/2N)$, which is right in open-source code.
2. Line 29: "Although there are some works^{18, 29} to for automated lens design" has grammar mistake.

Summary :

The authors have not adequately addressed the previous comments about tolerance analysis. There still exist some concerns about the evaluation of EDoF in the revised article. The authors need to make further revisions.

Reviewer #3 (Remarks to the Author):

The response to the reviewer provides by the author failed to answer my concern.

The lens design presented in the paper is very far from what could be acceptable in the field.

1) The authors didn't take into account of the real waveband responses. This response: 'This approach is reflected in commercial software like ZEMAX where three wavelengths are often enough to characterize RGB channels.' is very outdated, the true color needs wavelengths down to 425nm (to produce magenta). The color must be computed according to the sensor response, limiting the design to 486nm makes it very easy to design. Consequently this paper is far from reality and opens a doubt that dealing with real chromatic problems can be solved with the proposed approach.

2) The MTF is not good (as well as spot size). The pixel pitch of 2.65 μ m will imply a minimum MTF at 94lp/mm ($N_y/2$) of more than 50% (supp.note 4 shows a MTF about 30%). Consequently, again does the process can achieve real good performance or it is only starting point design that still needs intensive design).

3) Two main parameters in a cell phone camera are the CRA mismatch and the relative illumination, both are not discussed. Again does the technique is limited?

4) For the EDoF, the authors failed to explain the gain of the technique such as why we need 4p aspheric lens rather than just spherical lens with a phase mask.

5) The comment about LensNet is surprising. I did my own test (because it is available which is not the case for the technique claim to be better by the author, so we can't test it!!). I used 35 deg FOV and F/4, the output from LensNet was all a spherical lens with spot size of 15 μ m? Which is more than enough as a starting point.

6) Does the curriculum learning can infer spherical only design when it is not required to use aspheric? It is a very fundamental question which is not answering and well within the comments of the reviewer saying that the paper is lack of comparison. How curriculum

learning compared to LensNet for a 40 deg FOV, F3.0?

Finally, I think that the authors have in hand a very nice AI applications but the paper is misleading as they claim that it gives optimized design. So I suggest writing the paper as a starting point generator for spherical and aspheric lens compositions (as long as the technic can provide spherical design when it is enough).

Author Response to the Manuscript: Curriculum Learning for *ab initio* Deep Learned Refractive Optics

Xinge Yang¹, Qiang Fu¹, and Wolfgang Heidrich^{1,*}

¹King Abdullah University of Science and Technology, Saudi Arabia

*Corresponding author: wolfgang.heidrich@kaust.edu.sa

We sincerely thank all reviewers for their insightful comments and suggestions. We have carefully revised the manuscript in accordance with these comments. We acknowledge that there are numerous trade-offs and considerations in lens design, not all of which can be considered in a single article introducing a new methodology.

In this work we focus specifically on enabling **deep end-to-end design** of **computational** imaging systems, i.e. the joint design of optics and algorithms. This is exemplified by the extended depth-of-field application. To this end, we have devised a new method that can optimize substantially more complex optical systems than ever before.

We demonstrate the ability of the approach to navigate complicated design spaces by running stress tests on classical optical design problems (i.e. designs without image processing component), however this is **not** the focus or motivation for our work. We strongly believe that most future imaging systems will be computational, i.e. they will require a tight interaction between optics and algorithms, and so far we have been lacking the design tools for such systems. We believe our work is a major step forward in this direction, since it for the first time allows us to tackle optical systems of a practically relevant complexity. Below is a detailed, point-by-point response to the comments.

Response to Reviewer #1:

Overview:

The revised paper is significantly improved. The paper now shows results on designs that are significantly different between the starting design specs and the final ones.

The paper also addressed many of the minor comments I had. Overall, I am for accepting this manuscript.

[Thank you for your positive comments. We are glad that you are satisfied with our revisions.]

Concerns:

a) Reading the paper again, the paper would benefit from expanding upon the results in Figure 2 / Table 2. Figure 2 results look a lot like the bicubic phase plate [Cathey and Dowski, 1997]. Could the paper comment on differences between the two? For example, how close is the optimized mask to a bicubic one?

[Thank you. Figure 2.b demonstrates the height profile of the odd-polynomial term of the hybrid surface (Eq. 3). Unlike the classical bicubic phase plate [Cathey and Dowski, 1997], it provides finer control of the height profile due to the inclusion of higher-order terms. We have included the corresponding discussion and evaluation in the revised manuscript (lines 124 and Supplementary Note 5.4).]

How does the depth invariance change across the field? The paper claims in Line 117 that the cubic phase plate has limitations due to “optical aberrations vary substantially across the image plane in such systems”. Can this be established by expanding on the results in Table 2, which currently only shows a marginal difference between the cubic plate and the optimized one.

[Thank you. In line 117, our intention is to underscore the limitations of previous EDoF studies, rather than the cubic phase plate itself. These studies^{1,2} idealize optical lenses as thin lenses without optical aberrations, an assumption that is not applicable to highly constrained design spaces such as mobile device form factors, which exhibit significant optical aberrations with a strong off-axis component. This is the primary reason why EDoF is successfully used in microscopy, but has yet to find its way into other imaging systems. In our experiments, we design hybrid surfaces and the cubic plate in the presence of optical aberrations, jointly with the reconstruction network. To the best of our knowledge, this represents the first effort to design a highly aspherical EDoF lens in a cellphone lens form.

To compare the depth invariance across the field between the hybrid surface and conventional cubic plate, we have added an experiment in Supplementary Note 5.4. We split the reconstructed images into “edge” and “center” regions, and calculate the PSNR of reconstructed images. The results show that the hybrid surface has a better reconstruction performance at the edge region, which credits to the higher odd-polynomial terms in the hybrid surface.]

b) Also, a typo in Fig 2 caption: “depp” to “deep”.

[Thank you. We have fixed it in the revised manuscript.]

Response to Reviewer #2:

Overview:

After reading the reviews, response, and the revised paper, I appreciate the authors' efforts in rewriting the paper and addressing all the raised issues by the reviewers.

I support the authors' work on the optical optimization of compound high-aspheric lenses from random initio. With three optical regularizations, the authors deploy numerous automated design experiments and alleviate 40% of the final designs from the panic of self-intersection. This significantly decreases the reliance on a well-structured initial design in the optimization of compound complex lens group. I am particularly thankful for the authors' open-source code, as their approach has the potential to make a significant impact on the broader community.

[Thank you for your insightful comments. We believe that differentiable optical design will have a significant impact not only on lens design but also on a broader field.]

I agree with authors' discussion about ISP and chromatic PSF. However, when it comes to analyzing the effects of manufacturing and assembly errors, the authors mention fine-tuning based on perturbations but don't provide corresponding simulation experiment data to validate this point.

[Thank you. Sorry for the misleading response. At the moment we do not determine the exact manufacturing and assembly errors for each individual lens (commonly known as "tolerancing"). Instead we use the network model trained for the designed lens to infer various perturbed lenses. The corresponding simulation results are provided in Supplementary Note 5.2. We have revised the wording in the revised manuscript (Supplementary Note 5.2). The network fine-tuning process has been discussed in a previous paper³, and we have added the corresponding discussion and citation in the revised manuscript (Supplementary Note 5.2).]

[It is also worth pointing out that the differentiable framework has strong potential for addressing the tolerancing problem: since gradient back-propagation provides exact information about the level of sensitivity for each manufacturing and assembly parameter, it is in principle possible to track the manufacturability of the design in the optimization process, and to penalize designs that require overly tight tolerances. We believe this is a very fruitful avenue for further investigation, however it is beyond the scope of this work.]

Concerns:

1. I cannot find any implementation details about the cubic phase plate in the article or in the open-source code. It appears to be deployed at the entrance pupil, but I'm unclear about how it introduces phase to the rays in this work.

[Thank you. We will make all the code open-source once the manuscript is accepted. In our approach, instead manipulating the phase directly, we represent the geometry of the cubic (or higher order) surface, and ray-trace it, just like we ray-trace the aspherical elements.]

2. In Fig2.e, the imaging quality of classical lenses, as measured by the Raw PSNR, has degraded approximately 3db compared to the first version of the article.

[Thank you. We have adjusted the EDoF lens from a 6-element, 57.3° , F/3.2 lens (original submission) to a 4-element, 68.8° , F/2.1 lens (previous revision) in response to Reviewer #3's comments. Consequently, all the EDoF results are new and unrelated to those in the original submission. Typically, a lens with fewer elements, a wider FoV, and a larger aperture will have worse imaging performance and a lower PSNR score.]

3. A reference line for the diffraction limit needs to be included in the MTF chart in Fig. 2f.

[Thank you. We have added the diffraction limit to the MTF chart in Fig. 2f in the revised manuscript.]

4. In Fig. 2g, the simulated results of the classical lens at 10 cm and 10 m have noticeable artifacts that are distinct from typical degradation caused by optical aberrations.

[Thank you. In Fig. 2.g, we are demonstrating the images after network reconstruction, rather than the RAW simulations. The artifacts are caused by the networks, therefore they differ from common optical aberrations.]

5. Classical lens should be further optimized, or alternatively, patent lens with the similar specifications can be employed to fairly evaluate the results of deep learned large-aperture EDoF lens.

[Thank you. We have optimized the classical lens to the best of our capabilities. Additionally, to the best of our knowledge, we have not found any patented lens with identical specifications.]

Some minor points:

1. Equation 2: the last term should be $\sin(i \pi/2N)$, which is right in open-source code.

[Thank you. We have fixed it in the revised manuscript.]

2. Line 29: “Although there are some works^{18, 29} to for automated lens design” has grammar mistake.

[Thank you. We have fixed it in the revised manuscript.]

Summary:

The authors have not adequately addressed the previous comments about tolerance analysis. There still exist some concerns about the evaluation of EDoF in the revised article. The authors need to make further revisions.

Response to Reviewer #3:

Overview:

The response to the reviewer provided by the author failed to answer my concern. The lens design presented in the paper is very far from what could be acceptable in the field.

[Thank you. It is important to note that the focus of this manuscript is an automatic lens design approach, especially for End-to-End computational lens design, a task not feasible with conventional lens design. The aim of this work is not to single-handedly replace software like ZEMAX that has been developed and refined over decades, but instead to introduce a new methodology that enables new applications, specifically Deep Lens design, i.e. the joint end-to-end design of optics and image reconstruction algorithms.

As such the classical design tasks are to be understood as stress tests for the method – for example there is no good reason to initialize a system with flat surfaces instead of a legacy design. Furthermore, we have demonstrated an approach to automatically design highly-aspherical compound lenses (up to 6P) from scratch, which, to the best of our knowledge, has never been achieved before. Therefore, we do not believe it is a fair requirement for us to automatically design lenses with the same performance as commercial lenses.

There are also some aspects of lens design that cannot be covered in a single paper, but we believe these should not diminish the significance of our work.]

Concerns:

1) The authors didn't take into account of the real waveband responses. This response: 'This approach is reflected in commercial software like ZEMAX where three wavelengths are often enough to characterize RGB channels.' is very outdated, the true color needs wavelengths down to 425nm (to produce magenta). The color must be computed according to the sensor response, limiting the design to 486nm makes it very easy to design. Consequently this paper is far from reality and open a doubt that dealing with real chromatic problem can be solved with the proposed approach.

[Thank you. We agree with your comment "This response: 'This approach is reflected in commercial software like ZEMAX where three wavelengths are often enough to characterize RGB channels.' is very outdated".

However, we disagree with the comment that "limiting the design to 486nm makes it very easy to design. Consequently, this paper is far from reality." The challenge in designing a lens from scratch does not lie in the wavelength selection. Instead, the importance lies in how to avoid self-intersection and local minima.

We do not view the wavelength as a significant issue. In Fig. 1 below, we present the PSF at three different fields (from left to right: on-axis, moderate FoV, and full FoV), as well as the average RMS spot size (calculated on 16 fields from on-axis to full FoV) of the final designs at different wavelengths. The results show that the RMS spot size of the final designs does not vary significantly with the wavelength, indicating that our lens design successfully reduces chromatic aberrations across a broad wavelength range.

Moreover, it is trivial to adjust the code to consider more wavelengths during the design if and when necessary: in the code ("optics.py"), modify line 2387 from

```
for j, wv in enumerate(WAVE_RGB):  
to
```

```
for j, wv in range(400, 701, 20):
```

for example. The code will be able to optimize the lens on 16 distinct wavelengths.]

Wavelength (nm)	PSF (3 fields)	Avg. RMS spot size (μm, 16 fields)	Wavelength (nm)	PSF (3 fields)	Avg. RMS spot size (μm, 16 fields)
400		5.956282	560		5.956489
420		5.955958	580		5.955805
440		5.956217	600		5.955150
460		5.956279	620		5.955481
480		5.955532	640		5.955406
500		5.955806	660		5.956075
520		5.956006	680		5.956273
540		5.955616	700		5.955763

Figure 1. Avg RMS spot size of the final designs at different wavelengths. Avg RMS spot size of the final designs does not vary significantly with the wavelength.

2) The MTF is not good (as well as spot size). The pixel pitch of 2.65μm will imply a minimum MTF at 94lp/mm (Ny/2) of more than 50% (supp.note 4 shows a MTF about 30%). Consequently, again does the process can achieve real good performance or it is only starting point design that still need intensive design.

[Thank you. After continue optimizing at higher image resolution for 50k more iterations, the lens design converges to a better performance. The MTF at 94lp/mm is about 50%, as shown in Fig. 2. It is challenging to achieve the same performance as commercial lenses, but we believe that our method is a significant step forward in the field of lens design. We have revised the manuscript to include the updated results.]

Figure 2. Analysis of the final design. The MTF at 94lp/mm is about 50%.

3) Two main parameters in a cell phone camera is the CRA mismatch and the relative illumination, both are not discussed. Again does the technique is limited?

[Thank you. We believe that the CRA mismatch and relative illumination considerations are similar to the lens manufacturing and ISP discussion. While we agree that they are important, they are not the focus of this paper. For example, the CRA is only meaningful after coatings have been considered, which well beyond the scope of this work.]

4) For the EDoF, the authors failed to explain the gain of the technique such as why we need 4p aspheric lens rather than just spherical lens with a phase mask.

[Thank you. In response to your previous comments, we have simplified the lens structure by adjusting it from a 6 element (6P) lens to a 4P lens. This is already **significantly** more constrained than current generation cell phone camera lenses, which use up to 7P of highly aspherical designs to achieve the short total track length (TTL) required for a mobile device camera. It is highly unfair and unrealistic to ask our system to achieve similar optical performance with a much more constrained design space! Moreover, the primary challenge in automated lens design is to navigate the difficult non-convex optimization landscape. Testing the system on smaller design spaces does not provide additional insight on how well the system manages this challenge.]

[To summary, we have demonstrated automated designs in a complex design space – up to 6P of highly aspherical lenses, which is common in the realm of mobile phone cameras, and we have demonstrated a wide FoV and a compact structure (line 116). To the best of our knowledge, this is the first attempt at designing a highly aspherical, large FoV EDoF lens.]

5) The comment about LensNet is surprising. I did my own test (because it is available which is not the case for the technique claim to be better by the author, so we can't test it!). I used 35 deg FOV and F/4, the output from LensNet was all a spherical lens with spot size of 15um? Which is more than enough as a starting point.

[Thank you. We have provided the lens design code in the previous revision. We want to emphasize our advantages against LensNet⁴ again:

- In LensNet, you do not have complete control over the lenses you want to design, such as glass type, total thickness, and aperture position. You must further optimize the lenses in software like ZEMAX to meet practical requirements, which is why LensNet is referred to as a “starting point generator”. However, with DeepLens, you can have complete control over these parameters.
- LensNet requires a large number of successful existing lens designs for training. For certain design specifications without a reference, for example, an 80° FoV and F/2.0, LensNet fails. However, DeepLens does not require any reference lens designs, as it relies on optimization methods.
- LensNet cannot design aspherical lenses, while DeepLens can design both spherical and aspherical lenses, and it supports other surface types.
- LensNet cannot perform End-to-End lens design, while DeepLens can jointly optimize the lens with the network.

In summary, we do not think a comparison with LensNets is crucial, as it focuses on a very specific task: classical lens design for spherical surfaces, with existing successful lens design references. The methodologies of LensNet and DeepLens are fundamentally different. We believe that our method is more general and applicable to a broader range of tasks. Once again, the emphasis of our manuscript is on End-to-End lens design, which is not achievable with conventional methods.]

6) Does the curriculum learning can infer spherical only design when it is not required to use aspheric? It is a very fundamental question which is not answering and well within the comments of the reviewer saying that the paper is lack of comparison. How curriculum learning compared to LensNet for a 40 deg FOV, F3.0?

[Thank you. Our method is also capable of designing spherical-only lenses, as demonstrated in the attached **two videos**. We did not include these results in the manuscript because we chose to showcase the capability of our method through highly aspherical lenses. It is impossible to include all lens design cases in a single paper.

To meet your requirement, we have conducted a comparison with LensNet. Since you did not provide the lens data for us to compare with, we selected the top-right 3P design from LensNet, as seen in Fig. 3. The design results of DeepLens and LensNet are shown in Fig. 4. In terms of the RMS radius, the DeepLens result demonstrates a more balanced performance across the field compared to LensNet, which is typically more desirable in optical design. Corresponding **Zemax files** are provided in the supplementary material.]

LensNet: lens design starting point generator

Get started

Enter the effective focal length, f-number and half field of view desired for your lens design project. Our deep learning framework will infer a selection of lens designs tailored to those specifications.

Focal length (in mm)

F-number

Half field of view (in degrees)

Results

focal length: 25.0 mm, f-number: 3.0, half field of view: 20.0°

Figure 3. LesNet results for 40° FoV, F/3.0. The top-right 3P design is chosen for comparison.

DeepLens (ours)

Surface: IMA		Spot Diagram	
2024/2/6	Zemax	Zemax OpticStudio 19.4	
Units are μm . Legend items refer to Wavelengths			
Field : 1 2 3			
RMS radius : 12.370 18.200 17.699			
GEO radius : 20.056 52.180 71.187			
Scale bar : 200 Reference : Chief Ray			

LensNet

Surface: IMA		Spot Diagram	
2024/2/6	Zemax	Zemax OpticStudio 19.4	
Units are μm . Legend items refer to Wavelengths			
Field : 1 2 3			
RMS radius : 9.274 14.740 22.825			
GEO radius : 15.833 45.617 106.439			
Scale bar : 200 Reference : Chief Ray			

Figure 4. Comparison of the lens design results between DeepLens and LensNet. DeepLens result has a more balanced performance across the field compared to LensNet, which is typically more desired in optical design.

Summary:

Finally, I think that the authors have in hand a very nice AI applications but the paper is misleading as they claim that it gives optimized design. So I suggest writing the paper as a starting point generator for spherical and aspheric lens compositions (as long as the technic can provide spherical design when it is enough).

[Again we would like to emphasize that the primary focus of our work is to allow of joint optimization of optics and algorithms, a.k.a. deep end-to-end design. We are **not** targeting classical lens design, and only provide such examples as stress tests for our algorithm, in order to highlight its ability to cope with complicated design spaces.]

References

1. Dowski, E. R. & Cathey, W. T. Extended depth of field through wave-front coding. *Appl. Opt.* **34**, 1859–1866 (1995).
2. Sitzmann, V. *et al.* End-to-end optimization of optics and image processing for achromatic extended depth of field and super-resolution imaging. *ACM Trans. Graph.* **37**, DOI: [10.1145/3197517.3201333](https://doi.org/10.1145/3197517.3201333) (2018).
3. Chen, S. *et al.* Computational optics for mobile terminals in mass production. *IEEE Trans. Pattern Anal. Mach. Intell.* (2022).
4. Côté, G., Lalonde, J.-F. & Thibault, S. Deep learning-enabled framework for automatic lens design starting point generation. *Opt. Express.* **29**, 3841–3854 (2021).

REVIEWERS' COMMENTS:

Reviewer #1 (Remarks to the Author):

The authors have addressed my remarks satisfactorily.

The paper would benefit from having some of the discussion regards to Cathey and Dowski's cubic design moved to the main paper. Parts of Fig S21 should be incorporated in Fig 2.

Reviewer #2 (Remarks to the Author):

The response to the reviewers provided by the authors raises concerns about credibility of the optimized design results.

1) The authors have conducted a comparison with LensNet (in Fig 1.a) and provided the corresponding Zemax files in the supplementary material. But when open the 'Spherical lens design comparison (ours).zmx', there lies additional merit function(in Fig 1.b), which means the result is optimized by Zemax. While in 'Spherical lens design comparison (baseline).zmx', there is no such merit function(in Fig 1.c). So the comparison is not fair and it is hard to decide whether the result by DeepLens is credible.

2. Based on the above concern, I'm also having doubts about the result of 'Fig.S7' in the revised supplementary document, which indicate the current optimized design result (in Fig 2.a) is significantly improved over the previous design (in Fig 2.b). The authors imply 'After continue optimizing at higher image resolution for 50k more iterations, the lens design converges to a better performance.' It not convincing that such a boost can be achieved by continue optimizing at higher image resolution for more iterations.

Fig 1 and Fig 2 are attached.

[**Editorial note:** the mentioned attached figures are presented after the end of the next page.]

Reviewer #3 (Remarks to the Author):

General concerns :

I would like to say that this work is useful and high quality. I think that the answers are good, and the paper is now better.

However, the new responses to the reviewers provide by the author failed to answer the main concerns. First the comparison between DeepLens and LensNet shows that DeepLens is also a starting point generator as LensNet (I do agree that DeepLens can provide design starting point using aspheric surfaces but it is not something new). Secondly, the author already published a paper 'C. Wang, N. Chen and W. Heidrich, "dO: A Differentiable Engine for Deep Lens Design of Computational Imaging Systems," in IEEE Transactions on Computational Imaging, vol. 8, pp. 905-916, 2022, doi: 10.1109/TCI.2022.3212837.' on using deep learning to design computational imaging system, so it is not something new. The third aspect, the authors mentioned in the abstract: 'However, it has been limited to either simple optical systems consisting of a single element such as a diffractive optical element (DOE) or metalens, or the fine-tuning of compound lenses from good initial designs.' But this is not the reality, we can find several paper in vision that dealt with this, a recent one is, 'G. Côté, F. Mannan, S. Thibault, J. -F. Lalonde and F. Heide, "The Differentiable Lens: Compound Lens Search over Glass Surfaces and Materials for Object Detection," 2023 IEEE/CVF Conference on Computer Vision and Pattern Recognition (CVPR), Vancouver, BC, Canada, 2023, pp. 20803-20812, doi: 10.1109/CVPR52729.2023.01993.' This group have also published on computational imaging using single metalens but the above paper shows a end to end design including optics toward a computational task. Consequently, for the three points above, I can't recommend the publication of this paper.

I still think that modern tools (such as DeepLens) should use real detector data response within the lens design, all these optimizations are useless if the detector CRA mismatch as well as color. In vision conference, researcher include IQ tuning (ISP response) because it will affect the computational task and results very much.

Fig 1.a the comparison between two designs

Fig 1.b Zemax file of Spherical lens design comparison (ours)

Fig 1.c Zemax file of Spherical lens design comparison (baseline)

Fig 2.a current design version

Fig 2.b previous design version

Point-to-point Response to Manuscript NCOMMS-23-10836C-Z: Curriculum Learning for *ab initio* Deep Learned Refractive Optics

Xinge Yang¹, Qiang Fu¹, and Wolfgang Heidrich^{1,*}

¹King Abdullah University of Science and Technology, Saudi Arabia

1 Reviewer 1:

The authors have addressed my remarks satisfactorily. The paper would benefit from having some of the discussion regards to Cathey and Dowski's cubic design moved to the main paper. Parts of Fig S21 should be incorporated in Fig 2.

We thank the reviewer for the positive feedback. We have moved the material from the supplement to the main paper (Fig 2) as requested.

2 Reviewer 2:

We believe we have answered all the concerns raised by Reviewer 2 in the second round of review. The final review of Reviewer 2 is based on misunderstandings.

The authors have conducted a comparison with LensNet (in Fig 1.a) and provided the corresponding Zemax files in the supplementary material. But when open the 'Spherical lens design comparison (ours).zmx', there lies additional merit function(in Fig 1.b), which means the result is optimized by Zemax. While in 'Spherical lens design comparison (baseline).zmx', there is no such merit function(in Fig 1.c). So the comparison is not fair and it is hard to decide whether the result by DeepLens is credible.

We would like to stress that **all presented designs** were optimized **exclusively** using our approach. At no point was Zemax used for any of the actual design process. As evidence of this statement, we have provided the experimental code for reviewers to re-implement the results, and the original experimental logging information is provided in Fig. R3 to support our results.

Regarding the figure shown in the response letter, we would like to stress two things:

- The spherical lens design is a comparison experiment requested by Reviewer 3, and it is not part of the paper, since this example does not even show the capabilities of our work.
- the design was loaded into Zemax *after optimization by our system* for using the Zemax analysis tools to prepare a response to Reviewer 3. We did not perform any other steps in Zemax.

Based on the above concern, I'm also having doubts about the result of Fig.S7 in the revised supplementary document, which indicate the current optimized design result (in Fig2.a) is significantly improved over the previous design (in Fig2.b). The authors imply 'After continue optimizing at higher image resolution for 50k more iterations, the lens design converges to a better performance.' It not convincing that such a boost can be achieved by continue optimizing at higher image resolution for more iterations.

We have provided the experimental code for reviewers to re-implement the results, and the original experimental logging information is provided in Fig. R1 and R2 to support our results.

Our lens design code has a great optimization power because it utilizes back-propagated gradients for lens design. Highly-aspherical lens design is a non-convex optimization problem, therefore needs an intensive optimization process for convergence. As shown in Fig. R1, despite the loss function decreasing slowly, the lens design can still be improved by optimizing more. Therefore, a boost in lens quality can be achieved by optimizing more iterations and at high resolution.

3 Reviewer 3

We believe we have answered all the concerns and performed the requested evaluations raised by Reviewer 3 in the second round of review. Reviewer 3's final review has factual errors.

“First the comparison between DeepLens and LensNet shows that DeepLens is also a starting point generator as LensNet (I do agree that DeepLens can provide design starting point using aspheric surfaces but it is not something new).”

This has factual errors. We use **different methods** (curriculum learning without training data or human intervention) and **different functional conditions** (fully automated end-to-end aspherical lens design from scratch). Also, fully automated cellphone-form aspherical lens design is **much harder** than spherical lens design, especially with neural network representations for the optics, since the optical design space has much higher dimensionality, which makes it difficult to train a network to provide optical grade precision over the full design space. Certainly something like this **has never been demonstrated!**

“Secondly, the author already published a paper ‘C. Wang, N. Chen and W. Heidrich, ”dO: A Differentiable Engine for Deep Lens Design of Computational Imaging Systems,” in IEEE Transactions on Computational Imaging, vol. 8, pp. 905-916, 2022, doi: 10.1109/TCI.2022.3212837.’ on using deep learning to design computational imaging system, so it is not something new.”

This has factual errors. The *dO* paper proposed a basic differentiable engine for optical design, but it is **not able** to design a lens from scratch and **required optical engineers** to operate the optimization process. It has been used as the baseline in our experiments and a **comprehensive comparison has been provided** in the paper (lines 91 - 105 and Table 1) since the first round of revision.

“The third aspect, the authors mentioned in the abstract: ‘However, it has been limited to either simple optical systems consisting of a single element such as a diffractive optical element (DOE) or metalens, or the fine-tuning of compound lenses from good initial designs.’ But this is not the reality, we can find several paper in vision that dealt with this, a recent one is, ‘G. Côté, F. Mannan, S. Thibault, J. -F. Lalonde and F. Heide, ”The Differentiable Lens: Compound Lens Search over Glass Surfaces and Materials for Object Detection,” 2023 IEEE/CVF Conference on Computer Vision and Pattern Recognition (CVPR), Vancouver, BC, Canada, 2023, pp. 20803-20812, doi: 10.1109/CVPR52729.2023.01993.”’

This has factual errors. First, Côté et al.'s CVPR paper can only perform end-to-end optical design **with successful starting points**, which has been mentioned in the sentence “*or the fine-tuning of compound lenses from good initial designs*”. Second, Côté et al's CVPR paper (published on 6/20/2023) is **three months later** than our first submission (submitted on 03/29/2023). Third, Côté et al's CVPR paper solves the problem of object detection while we propose an EDoF lens with a cellphone form factor (highly-aspherical and large field-of-view), which is **fundamentally different**.

We believe the authors of Côté et al. would be the first to recognize and acknowledge these shortcomings, and encourage asking them for their input in case there is any doubt about our statement.

I still think that modern tools (such as DeepLens) should use real detector data response within the lens design, all these optimizations are useless if the detector CRA mismatch as well as color. In vision conference, researcher include IQ tuning (ISP response) because it will affect the computational task and results very much.

We have tried our best to address the majority of considerations raised, and we believe it is **impossible** to cover all aspects in a single paper. To our knowledge, **no existing paper** covers all the considerations. For example works on end-to-end designed metasurfaces or related technologies in Nature journals never include a full ISP pipeline, so we believe that this request is holding us to an impossible standard that no-one else is asked to satisfy.

In summary, the biggest contribution of our work is that we can perform a fully automated end-to-end lens design from scratch, while others' work can not. Moreover, our lens optimization can function without relying on any existing human knowledge, which is fundamentally different from other works.

```
iter18900.png AutoLens.32171375.err x
AutoLens.32171375.err
1 ERROR: Unable to locate a modulefile for 'cuda'
2 2024-02-06 11:23:42; [NF0;lr:[1e-05, 1e-05, 0.1, 1e-05], decay:0.1, iteration
3
4 Progress: 0% | 0/20001 [00:00<7, 7it/s, rms=0]
5 Progress: 0% | 0/20001 [00:00<7, 7it/s, rms=0.00796]
6 Progress: 0% | 1/20001 [00:00<45:46:50, 8.24s/it, rms=0.00796]
7 Progress: 0% | 1/20001 [00:09<45:46:50, 8.24s/it, rms=0.00796]
8 Progress: 0% | 2/20001 [00:09<24:03:33, 4.33s/it, rms=0.00796]
9 Progress: 0% | 2/20001 [00:11<24:03:33, 4.33s/it, rms=0.00796]
10 Progress: 0% | 3/20001 [00:11<17:05:57, 3.08s/it, rms=0.00796]
11 Progress: 0% | 3/20001 [00:13<17:05:57, 3.08s/it, rms=0.00795]
12 Progress: 0% | 4/20001 [00:13<14:01:56, 2.53s/it, rms=0.00795]
13 Progress: 0% | 4/20001 [00:14<14:01:56, 2.53s/it, rms=0.00795]
14 Progress: 0% | 5/20001 [00:14<12:10:21, 2.19s/it, rms=0.00795]
15 Progress: 0% | 5/20001 [00:16<12:10:21, 2.19s/it, rms=0.00794]
16 Progress: 0% | 6/20001 [00:16<11:03:53, 1.99s/it, rms=0.00794]
17 Progress: 0% | 6/20001 [00:17<11:03:53, 1.99s/it, rms=0.00794]
18 Progress: 0% | 7/20001 [00:17<10:20:59, 1.86s/it, rms=0.00794]
19 Progress: 0% | 7/20001 [00:19<10:20:59, 1.86s/it, rms=0.00795]
20 Progress: 0% | 8/20001 [00:19<10:11:52, 1.84s/it, rms=0.00795]
21 Progress: 0% | 8/20001 [00:21<10:11:52, 1.84s/it, rms=0.00795]

37835 Progress: 95% | 18915/20001 [8:59:57<29:25, 1.63s/it, rms=0.00548]
37836 Progress: 95% | 18916/20001 [8:59:57<29:00, 1.60s/it, rms=0.00548]
37837 Progress: 95% | 18916/20001 [8:59:59<29:00, 1.60s/it, rms=0.00548]
37838 Progress: 95% | 18917/20001 [8:59:59<28:38, 1.59s/it, rms=0.00548]
37839 Progress: 95% | 18917/20001 [9:00:00<28:38, 1.59s/it, rms=0.00548]
37840 Progress: 95% | 18918/20001 [9:00:00<28:45, 1.59s/it, rms=0.00548]
37841 Progress: 95% | 18918/20001 [9:00:02<28:45, 1.59s/it, rms=0.00548]
37842 Progress: 95% | 18919/20001 [9:00:02<28:50, 1.60s/it, rms=0.00548]
37843 Progress: 95% | 18919/20001 [9:00:04<28:50, 1.60s/it, rms=0.00548]
37844 Progress: 95% | 18920/20001 [9:00:04<28:29, 1.58s/it, rms=0.00548]
37845 Progress: 95% | 18920/20001 [9:00:05<28:29, 1.58s/it, rms=0.00548]
37846 Progress: 95% | 18921/20001 [9:00:05<28:36, 1.59s/it, rms=0.00548]
37847 Progress: 95% | 18921/20001 [9:00:07<28:36, 1.59s/it, rms=0.00548]
37848 Progress: 95% | 18922/20001 [9:00:07<28:21, 1.58s/it, rms=0.00548]
37849 Progress: 95% | 18922/20001 [9:00:08<28:21, 1.58s/it, rms=0.00549]
37850 Progress: 95% | 18923/20001 [9:00:08<28:08, 1.57s/it, rms=0.00549]
37851 Progress: 95% | 18923/20001 [9:00:10<28:08, 1.57s/it, rms=0.00549]
37852 Progress: 95% | 18924/20001 [9:00:10<28:20, 1.58s/it, rms=0.00549]
37853 Progress: 95% | 18924/20001 [9:00:12<28:20, 1.58s/it, rms=0.00549]
37854 Progress: 95% | 18925/20001 [9:00:12<28:39, 1.60s/it, rms=0.00549]
37855 Progress: 95% | 18925/20001 [9:00:13<28:39, 1.60s/it, rms=0.00549]
37856 Progress: 95% | 18926/20001 [9:00:13<28:44, 1.60s/it, rms=0.00549]
37857 Progress: 95% | 18926/20001 [9:00:15<28:44, 1.60s/it, rms=0.00549]
37858 Progress: 95% | 18927/20001 [9:00:15<28:45, 1.61s/it, rms=0.00549]
06T20:23:58 DUE TO TIME LIMIT ***
```

Lens optimization logging output

```
iter18900.png AutoLens.32171375.out x
AutoLens.32171375.out
1 On-axis RMS radius: 8.786um, Off-axis RMS radius: 14.488um, Avg RMS spot size (radius): 8.54um.
2 On-axis RMS radius: 8.793um, Off-axis RMS radius: 14.308um, Avg RMS spot size (radius): 8.526um.
3 On-axis RMS radius: 8.536um, Off-axis RMS radius: 14.478um, Avg RMS spot size (radius): 8.518um.
4 On-axis RMS radius: 8.456um, Off-axis RMS radius: 14.548um, Avg RMS spot size (radius): 8.506um.
5 On-axis RMS radius: 8.422um, Off-axis RMS radius: 14.161um, Avg RMS spot size (radius): 8.494um.
6 On-axis RMS radius: 8.293um, Off-axis RMS radius: 14.242um, Avg RMS spot size (radius): 8.494um.
7 On-axis RMS radius: 8.309um, Off-axis RMS radius: 14.303um, Avg RMS spot size (radius): 8.458um.
8 On-axis RMS radius: 8.181um, Off-axis RMS radius: 14.17um, Avg RMS spot size (radius): 8.462um.
9 On-axis RMS radius: 8.114um, Off-axis RMS radius: 14.255um, Avg RMS spot size (radius): 8.452um.
10 On-axis RMS radius: 8.146um, Off-axis RMS radius: 14.159um, Avg RMS spot size (radius): 8.428um.
11 On-axis RMS radius: 7.989um, Off-axis RMS radius: 14.205um, Avg RMS spot size (radius): 8.423um.
12 On-axis RMS radius: 8.051um, Off-axis RMS radius: 14.155um, Avg RMS spot size (radius): 8.385um.
13 On-axis RMS radius: 7.953um, Off-axis RMS radius: 14.226um, Avg RMS spot size (radius): 8.373um.
14 On-axis RMS radius: 8.063um, Off-axis RMS radius: 13.956um, Avg RMS spot size (radius): 8.339um.
15 On-axis RMS radius: 7.885um, Off-axis RMS radius: 13.976um, Avg RMS spot size (radius): 8.334um.
16 On-axis RMS radius: 8.054um, Off-axis RMS radius: 14.0um, Avg RMS spot size (radius): 8.328um.
17 On-axis RMS radius: 7.902um, Off-axis RMS radius: 13.998um, Avg RMS spot size (radius): 8.272um.
18 On-axis RMS radius: 7.877um, Off-axis RMS radius: 13.985um, Avg RMS spot size (radius): 8.261um.
19 On-axis RMS radius: 7.955um, Off-axis RMS radius: 13.83um, Avg RMS spot size (radius): 8.252um.
20 On-axis RMS radius: 7.919um, Off-axis RMS radius: 13.978um, Avg RMS spot size (radius): 8.226um.
21 On-axis RMS radius: 7.839um, Off-axis RMS radius: 13.6um, Avg RMS spot size (radius): 8.189um.
22 On-axis RMS radius: 7.956um, Off-axis RMS radius: 13.597um, Avg RMS spot size (radius): 8.142um.
23 On-axis RMS radius: 7.833um, Off-axis RMS radius: 13.499um, Avg RMS spot size (radius): 8.133um.
24 On-axis RMS radius: 7.941um, Off-axis RMS radius: 13.246um, Avg RMS spot size (radius): 8.097um.
25 On-axis RMS radius: 7.894um, Off-axis RMS radius: 13.309um, Avg RMS spot size (radius): 8.085um.

143 On-axis RMS radius: 7.16um, Off-axis RMS radius: 10.591um, Avg RMS spot size (radius): 6.33um.
144 On-axis RMS radius: 7.196um, Off-axis RMS radius: 10.478um, Avg RMS spot size (radius): 6.328um.
145 On-axis RMS radius: 7.152um, Off-axis RMS radius: 10.658um, Avg RMS spot size (radius): 6.325um.
146 On-axis RMS radius: 7.176um, Off-axis RMS radius: 10.582um, Avg RMS spot size (radius): 6.319um.
147 On-axis RMS radius: 7.154um, Off-axis RMS radius: 10.752um, Avg RMS spot size (radius): 6.292um.
148 On-axis RMS radius: 7.125um, Off-axis RMS radius: 10.706um, Avg RMS spot size (radius): 6.296um.
149 On-axis RMS radius: 7.141um, Off-axis RMS radius: 10.501um, Avg RMS spot size (radius): 6.294um.
150 On-axis RMS radius: 7.148um, Off-axis RMS radius: 10.54um, Avg RMS spot size (radius): 6.292um.
151 On-axis RMS radius: 7.147um, Off-axis RMS radius: 10.468um, Avg RMS spot size (radius): 6.29um.
152 On-axis RMS radius: 7.132um, Off-axis RMS radius: 10.629um, Avg RMS spot size (radius): 6.265um.
153 On-axis RMS radius: 7.123um, Off-axis RMS radius: 10.508um, Avg RMS spot size (radius): 6.268um.
154 On-axis RMS radius: 7.106um, Off-axis RMS radius: 10.612um, Avg RMS spot size (radius): 6.268um.
155 On-axis RMS radius: 7.112um, Off-axis RMS radius: 10.696um, Avg RMS spot size (radius): 6.267um.
156 On-axis RMS radius: 7.105um, Off-axis RMS radius: 10.525um, Avg RMS spot size (radius): 6.262um.
157 On-axis RMS radius: 7.13um, Off-axis RMS radius: 10.603um, Avg RMS spot size (radius): 6.239um.
158 On-axis RMS radius: 7.125um, Off-axis RMS radius: 10.55um, Avg RMS spot size (radius): 6.239um.
159 On-axis RMS radius: 7.12um, Off-axis RMS radius: 10.552um, Avg RMS spot size (radius): 6.241um.
160 On-axis RMS radius: 7.108um, Off-axis RMS radius: 10.492um, Avg RMS spot size (radius): 6.242um.
161 On-axis RMS radius: 7.004um, Off-axis RMS radius: 10.508um, Avg RMS spot size (radius): 6.236um.
162 On-axis RMS radius: 7.092um, Off-axis RMS radius: 10.601um, Avg RMS spot size (radius): 6.213um.
163 On-axis RMS radius: 7.081um, Off-axis RMS radius: 10.601um, Avg RMS spot size (radius): 6.215um.
164 On-axis RMS radius: 7.049um, Off-axis RMS radius: 10.615um, Avg RMS spot size (radius): 6.213um.
165 On-axis RMS radius: 7.048um, Off-axis RMS radius: 10.53um, Avg RMS spot size (radius): 6.211um.
166 On-axis RMS radius: 7.067um, Off-axis RMS radius: 10.582um, Avg RMS spot size (radius): 6.211um.
167 On-axis RMS radius: 7.072um, Off-axis RMS radius: 10.629um, Avg RMS spot size (radius): 6.189um.
168 On-axis RMS radius: 7.035um, Off-axis RMS radius: 10.615um, Avg RMS spot size (radius): 6.191um.
169
```

Lens optimization logging output

Figure R1. The original experimental logging information for the aspherical lens shown in Fig.S7. The lens was optimized in an experiment on 02/06/2024. As the optimization going on, both the loss function and the RMS spot size decreases. The experiment ended with a timing issue, and we selected the last output (iteration 18900) as the final result shown in Fig.S7.

0206-112342-DesignLensRMS-6P-asphe

- iter0_psf20000mm.png
- iter0.png
- iter0.txt
- iter100_psf20000mm.png
- iter100.png
- iter100.txt
- iter200_psf20000mm.png
- iter200.png
- iter200.txt
- iter300_psf20000mm.png
- iter300.png
- iter300.txt
- iter400_psf20000mm.png
- iter400.png
- iter400.txt
- iter500_psf20000mm.png
- iter500.png
- iter500.txt
- iter600_psf20000mm.png
- iter600.png
- iter600.txt
- iter700_psf20000mm.png
- iter700.png
- iter700.txt
- iter800_psf20000mm.png

- iter18300.txt
- iter18400_psf20000mm.png
- iter18400.png
- iter18400.txt
- iter18500_psf20000mm.png
- iter18500.png
- iter18500.txt
- iter18600_psf20000mm.png
- iter18600.png
- iter18600.txt
- iter18700_psf20000mm.png
- iter18700.png
- iter18700.txt
- iter18800_psf20000mm.png
- iter18800.png
- iter18800.txt
- iter18900_psf20000mm.png
- iter18900.png
- iter18900.txt
- output.log

```

results > 0206-112342-DesignLensRMS-6P-asphe > iter18900.txt
1 optimized lens file.
2 type distance roc diameter material
3 0 0 0 0 AIR
4 S 0.0000 0.0000000 2.18 air 0.00
5 S 0.0212 0.03914398 3.09 coc 21.17 1.018628e-01 -8.684750e-03 -4.611520e-03 -7.929758e-04 -1.029136e-04 -1.179749e-05
6 S 0.7698 -0.00164641 3.44 air 2.95 -7.214873e-02 -2.636273e-03 -1.944682e-03 -2.881352e-04 -2.455179e-05 -8.686996e-07
7 S 0.1660 -0.01702422 3.72 ps 14.75 -8.112657e-02 2.650892e-03 2.527483e-03 3.431476e-04 2.736465e-05 7.491061e-07
8 S 0.4820 0.00796670 4.02 air 7.86 5.848620e-02 -5.603598e-03 -1.984355e-03 -3.076744e-04 -3.419986e-05 -3.013467e-06
9 S 0.1941 0.00915996 4.24 coc -12.15 2.786103e-02 1.511312e-03 -6.370649e-04 -1.334850e-04 -2.360612e-05 -3.668728e-06
10 S 0.6665 -0.01163926 4.41 air 37.39 -4.787914e-02 -5.482937e-03 5.193869e-04 1.209104e-04 1.709787e-05 2.884847e-06
11 S 0.2848 0.00321182 4.76 ps 3.10 2.298320e-02 -1.242724e-03 -5.231182e-04 -1.069096e-04 -1.543843e-05 -1.640627e-06
12 S 0.8618 -0.02497874 4.91 air 7.75 -8.083490e-02 3.590619e-03 6.461686e-04 5.692690e-05 4.513090e-06 1.628945e-07
13 S 0.6222 0.01806958 5.40 okp4 -39.03 6.382249e-02 -9.967105e-03 -8.356384e-04 -7.687697e-06 4.346374e-06 5.636686e-07
14 S 0.6016 -0.00055203 5.74 air 3.47 3.088906e-03 2.243520e-04 -4.045716e-04 -7.404666e-05 -4.760906e-06 4.209208e-07
15 S 0.5785 -0.02659353 5.94 pc -37.05 -1.160145e-01 2.036841e-03 3.560550e-04 1.290358e-05 -2.266536e-06 -6.655992e-07
16 S 0.4020 0.02752691 7.15 air -58.15 7.248812e-02 -7.317441e-03 -8.405925e-05 2.419948e-05 1.882054e-06 -2.325218e-07
17 I 1.243 0.0 7.66 air

```

Lens optimization output files

Lens optimization final output

Lens Data

Update: All Windows

Surface 0 Properties Configuration 1/1

Surface	Type	Radius	Thickness	Material	Coat	Clear Sel	Chi	Mech Sen	Conic	TCE x 1E	Par 1 (unused)	Par 2 (unused)	Par 3 (unused)	Par 4 (unused)	Par 5 (unused)
0	OB	Standard	Infinity	Infinity			Infini...	0.0.	Infinity	0.000	0.000				
1	ST	Standard	Infinity	0.021			1.090	0.0.	1.090	0.000	0.000				
2	(a) Even Asphere	25.547	0.770	COC			1.545 U	0.0.	1.720	21.170	-	0.102	-8.685E-03	-4.612E-03	-7.930E-04
3	(a) Even Asphere	-607.3...	0.166				1.720 U	0.0.	1.720	2.950	0.000	-0.072	-2.636E-03	-1.945E-03	-2.881E-04
4	(a) Even Asphere	-58.741	0.482	POLYS...			1.860 U	0.0.	2.010	14.750	-	-0.081	2.651E-03	2.527E-03	3.431E-04
5	(a) Even Asphere	125.522	0.194				2.010 U	0.0.	2.010	7.860	0.000	0.058	-5.604E-03	-1.984E-03	-3.077E-04
6	(a) Even Asphere	109.170	0.666	COC			2.120 U	0.0.	2.205	-12.150	-	0.028	1.511E-03	-6.371E-04	-1.335E-04
7	(a) Even Asphere	-85.918	0.285				2.205 U	0.0.	2.205	37.390	0.000	-0.048	-5.483E-03	5.194E-04	1.209E-04
8	(a) Even Asphere	311.352	0.862	POLYS...			2.380 U	0.0.	2.455	3.100	-	0.023	-1.243E-03	-5.231E-04	-1.069E-04
9	(a) Even Asphere	-40.046	0.622				2.455 U	0.0.	2.455	7.750	0.000	-0.081	3.591E-03	6.462E-04	5.693E-05
10	(a) Even Asphere	55.340	0.602	OKP4			2.700 U	0.0.	2.870	-39.030	-	0.064	-9.967E-03	-8.356E-04	-7.688E-06
11	(a) Even Asphere	-1811....	0.578				2.870 U	0.0.	2.870	3.470	0.000	3.089E-03	2.244E-04	-4.046E-04	-7.404E-05
12	(a) Even Asphere	-37.602	0.402	POLYC...			2.970 U	0.0.	3.575	-37.050	-	-0.116	2.037E-03	3.561E-04	1.290E-05
13	(a) Even Asphere	36.328	1.243				3.575 U	0.0.	3.575	-58.150	0.000	0.072	-7.317E-03	-8.406E-05	2.420E-05
14	IM	Standard	Infinity	-			3.798	0.0.	3.798	0.000	0.000				1.882E-06

Standard - Automatic

Line Thickness

Standard - Automatic

Figure R2. The original experimental output files for the aspherical lens shown in Fig.S7. The iteration 18900 was selected as the final result. We then loaded the lens data into Zemax for evaluation. The spot size in Zemax is slightly larger than that in our code, which is reasonable. The corresponding spot diagram and MTF curve are used in Fig.S7.

Figure R3. The original experimental output files for the 3P spherical lens. Iteration 20000 was selected and we loaded it into Zemax for final evaluation. It also important to note that the spherical lens design experiment is only a request by Reviewer 3, which is not the focus or an application of our paper.

REVIEWERS' COMMENTS

Reviewer #2 (Remarks to the Author):

The authors have addressed my concerns. I would like to say that the work is useful and optical regularization introduced in the paper does effectively prevent the optimization process from self-intersecting geometries or aggressive aspheric shapes.

However, the MTF plot of Figure R2 in response letter and Fig. S7 in supplementary file are still misleading. The MTF curves may give other researchers the wrong impression that the design achieves satisfactory optimization. In fact, the MTF curves presented in the paper are based on misunderstandings.

The followed comments are based on the Lens design aspherical file (ZMX).

1) It is common to use FFT MTF in ZEMAX to evaluate optical systems. With proper sampling setup (Fig. 1(a)), the MTF curve is showed in Fig. 1(d) which is much worse than presented in the paper.

2) In my point of view, the authors may use Huygens MTF in ZEMAX to generate the curves in Figure R2 and Fig. S7. Huygens MTF can be used to evaluate the optical systems properly unless the sampling rate is high enough to present the correct curves. The MTF curve in Fig. 1(e) is similar to Figure R2 and Fig. S7 with default pupil and image sampling (Fig. 1(b)). But the default sampling is not high enough to give correct result. As in Fig. 1(c), the sampling is set higher and the corresponding MTF curve in Fig. 1(f) is much similar with Fig. 1(d).

Please find Fig. 1 from attached.

Point-to-point Response to Manuscript NCOMMS-23-10836C-Z: Curriculum Learning for *ab initio* Deep Learned Refractive Optics

Xinge Yang¹, Qiang Fu¹, and Wolfgang Heidrich^{1,*}

¹King Abdullah University of Science and Technology, Saudi Arabia

1 Reviewer 2:

The authors have addressed my concerns. I would like to say that the work is useful and optical regularization introduced in the paper does effectively prevent the optimization process from self-intersecting geometries or aggressive aspheric shapes.

However, the MTF plot of Figure R2 in response letter and Fig. S7 in supplementary file are still misleading. The MTF curves may give other researchers the wrong impression that the design achieves satisfactory optimization. In fact, the MTF curves presented in the paper are based on misunderstandings.

The followed comments are based on the Lens design aspherical file (ZMX).

1) It is common to use FFT MTF in ZEMAX to evaluate optical systems. With proper sampling setup (Fig. 1(a)), the MTF curve is showed in Fig. 1(d) which is much worse than presented in the paper.

2) In my point of view, the authors may use Huygens MTF in ZEMAX to generate the curves in Figure R2 and Fig. S7. Huygens MTF can be used to evaluate the optical systems properly unless the sampling rate is high enough to present the correct curves. The MTF curve in Fig. 1(e) is similar to Figure R2 and Fig. S7 with default pupil and image sampling (Fig. 1(b)). But the default sampling is not high enough to give correct result. As in Fig. 1(c), the sampling is set higher and the corresponding MTF curve in Fig. 1(f) is much similar with Fig. 1(d). Please find Fig. 1 from attached.

Thanks for the positive feedback. We apologize for the confusion caused by the MTF plots. We have revised the manuscript to use the FFT MTF plots.

Polychromatic Diffraction MTF	
2024/6/9 Data for 0.4861 to 0.6563 μm . Surface: Image Legend items refer to Field positions	Zemax Zemax OpticStudio 19.4 asphe_80deg_f2_design2.ZMX Configuration 1 of 1

Figure R1. We have revised the MTF plots to use the FFT MTF plots.